# Measuring the sequence-affinity landscape of antibodies with massively parallel titration curves

Rhys M Adams[1,2‡], Thierry Mora[3*†], Aleksandra M Walczak[1*†], Justin B Kinney[2*†]

[1]Laboratoire de Physique Théorique, UMR8549, CNRS, École Normale Supérieure, Paris, France; [2]Simons Center for Quantitative Biology, Cold Spring Harbor Laboratory, Cold Spring Harbor, United States; [3]Laboratoire de Physique Statistique, UMR8550, CNRS, École Normale Supérieure, Paris, France

**Abstract** Despite the central role that antibodies play in the adaptive immune system and in biotechnology, much remains unknown about the quantitative relationship between an antibody's amino acid sequence and its antigen binding affinity. Here we describe a new experimental approach, called Tite-Seq, that is capable of measuring binding titration curves and corresponding affinities for thousands of variant antibodies in parallel. The measurement of titration curves eliminates the confounding effects of antibody expression and stability that arise in standard deep mutational scanning assays. We demonstrate Tite-Seq on the CDR1H and CDR3H regions of a well-studied scFv antibody. Our data shed light on the structural basis for antigen binding affinity and suggests a role for secondary CDR loops in establishing antibody stability. Tite-Seq fills a large gap in the ability to measure critical aspects of the adaptive immune system, and can be readily used for studying sequence-affinity landscapes in other protein systems.

**\*For correspondence:** tmora@lps. ens.fr (TM); awalczak@lpt.ens.fr (AMW); jkinney@cshl.edu (JBK)

[†]These authors contributed equally to this work

**Present address:** [‡]Francis Crick Institute, London, United Kingdom

**Competing interests:** The authors declare that no competing interests exist.

## Introduction

During an infection, the immune system must recognize and neutralize invading pathogens. B-cells contribute to immune defense by producing antibodies, proteins that bind specifically to foreign antigens. The astonishing capability of antibodies to recognize virtually any foreign molecule has been repurposed by scientists in a wide variety of experimental techniques (immunofluorescence, western blots, ELISA, ChIP-Seq, etc.). Antibody-based therapeutic drugs have also been developed for treating many different diseases, including cancer (*Chan and Carter, 2010*).

Much is known about the qualitative mechanisms of antibody generation and function (*Murphy et al., 2008*). The antigenic specificity of antibodies in humans, mice, and most jawed vertebrates is primarily governed by six complementarity determining regions (CDRs), each roughly 10 amino acids (aa) long. Three CDRs (denoted CDR1H, CDR2H, and CDR3H) are located on the antibody heavy chain, and three are on the light chain. During B-cell differentiation, these six sequences are randomized through V(D)J recombination, then selected for functionality as well as against the ability to recognize host antigens. Upon participation in an immune response, CDR regions can further undergo somatic hypermutation and selection, yielding higher-affinity antibodies for specific antigens. Among the CDRs, CDR3H is the most highly variable and typically contributes the most to antigen specificity; less clear are the functional roles of the other CDRs, which often do not interact with the target antigen directly.

Many high-throughput techniques, including phage display (*Smith, 1985*; *Vaughan et al., 1996*; *Schirrmann et al., 2011*), ribosome display (*Fujino et al., 2012*), yeast display (*Boder and Wittrup, 1997*; *Gai and Wittrup, 2007*), and mammalian cell display (*Forsyth et al., 2013*), have been

**eLife digest** Antibodies are proteins produced by cells of the immune system to tag or neutralize potential threats to the body, such as foreign substances and disease-causing microbes. Antibodies do this by binding to target molecules called antigens. An antibody's ability to bind to an antigen depends on the sequence of amino acids – the building blocks of proteins – that make up the antibody. Through a process that randomizes this sequence of amino acids, the immune system generates a vast pool of antibodies that are able to target almost any foreign antigen that exists in nature.

Currently, little is understood about how the sequence of amino acids in an antibody determines how strongly that antibody binds to its antigen target – a property referred to as the antibody's binding affinity. Answering this fundamental question requires techniques that can measure the affinities of many different antibodies at the same time. However, previous high-throughput methods have been unable to provide quantitative measurements of binding affinities. These kinds of measurements are difficult because an antibody's amino acid sequence governs more than just binding affinity: it also affects how easy it is to produce that antibody, and what fraction of antibody molecules work properly.

Adams et al. now describe a new method, named "Tite-Seq", that overcomes these issues. First, thousands of different antibodies are displayed on the surface of yeast cells, with each cell carrying a single kind of antibody. These cells are then incubated with fluorescently labeled antigen at a wide range of different concentrations. Next, the yeast cells are sorted based on how brightly they glow; brighter cells have more antigen bound to them, and so it is possible to calculate how much of the antigen is bound to each kind of antibody at each concentration. Plotting these data provides a "binding curve" for each antibody, which is then used to read off the antibody's binding affinity in a way that is not affected by the factors that have plagued other high-throughput methods.

Tite-Seq is thus able to measure the binding affinities for thousands of different antibodies at the same time. This will potentially allow researchers to address many fundamental and yet unanswered questions about how the immune system works. Tite-Seq can also be used to measure how amino acid sequence affects the binding affinity of proteins other than antibodies.

developed for optimizing antibodies ex vivo. Advances in DNA sequencing technology have also made it possible to effectively monitor both antibody and T-cell receptor diversity within immune repertoires, e.g. in healthy individuals (*Boyd et al., 2009*; *Weinstein et al., 2009*; *Robins et al., 2009*, *2010*; *Mora et al., 2010*; *Venturi et al., 2011*; *Murugan et al., 2012*; *Zvyagin et al., 2014*; *Elhanati et al., 2014*; *Qi et al., 2014*; *Thomas et al., 2014*; *Elhanati et al., 2015*), in specific tissues (*Madi et al., 2014*), in individuals with diseases (*Parameswaran et al., 2013*) or following vaccination (*Jiang et al., 2013*; *Vollmers et al., 2013*; *Laserson et al., 2014*; *Galson et al., 2014*; *Wang et al., 2015*). Yet many questions remain about basic aspects of the quantitative relationship between antibody sequence and antigen binding affinity. How many different antibodies will bind a given antigen with specified affinity? How large of a role do epistatic interactions between amino acid positions within the CDRs have on antigen binding affinity? How is this sequence-affinity landscape navigated by the V(D)J recombination process, or by somatic hypermutation? Answering these and related questions is likely to prove critical for developing a systems-level understanding of the adaptive immune system, as well as for using antibody repertoire sequencing to diagnose and monitor disease.

Recently developed 'deep mutational scanning' (DMS) assays (*Fowler and Fields, 2014*) provide one potential method for measuring binding affinities with high enough throughput to effectively explore antibody sequence-affinity landscapes. In DMS experiments, one begins with a library of variants of a specific protein. Proteins that have high levels of a particular activity of interest are then enriched via one or more rounds of selection, which can be carried out in a variety of ways. The set of enriched sequences is then compared to the initial library, and protein sequences (or mutations within these sequences) are scored according to how much this enrichment procedure increases their prevalence.

Multiple DMS assays have been described for investigating protein-ligand binding affinity. But no DMS assay has yet been shown to provide absolute quantitative binding affinity measurements, i.e., dissociation constants in molar units. For example, one of the first DMS experiments (*Fowler et al., 2010*) used phage display technology to measure how mutations in a WW domain affect the affinity of this domain for its peptide ligand. These data were sufficient to compute enrichment ratios and corresponding sequence logos, but they did not yield quantitative affinities. Analogous experiments have since been performed on antibodies using yeast display (*Reich et al., 2015*; *Kowalsky et al., 2015*) and mammalian cell display (*Forsyth et al., 2013*). Yeast-display-based DMS assays have also proven particularly useful for mapping protein epitopes that are targeted by specific antibodies of interest (*Kowalsky et al., 2015*; *Doolan and Colby, 2015*; *Van Blarcom et al., 2015*). Still, none of these approaches provides quantitative affinity values. SORTCERY (*Reich et al., 2015*, ), a DMS assay that combines yeast display and quantitative modeling, has been shown to provide approximate rank-order values for the affinity of a specific protein for short unstructured peptides of varying sequence. Determining quantitative affinities from SORTCERY data, however, requires separate low-throughput calibration measurements (*Reich et al., 2014*). Moreover, it is unclear how well SORT-CERY, if applied to a library of folded proteins rather than unstructured peptides, can distinguish sequence-dependence effects on affinity from sequence-dependent effects on protein expression and stability. Other recent work has described a DMS assay, again based on yeast display, for measuring fold-changes in affinity relative to a reference protein (*Kowalsky and Whitehead, 2016*). This method, however, does not provide absolute values for dissociation constants, is vulnerable to the confounding effects of sequence-dependent expression and protein stability, and was observed to have only a 10-fold dynamic range.

To enable massively parallel measurements of absolute binding affinities for antibodies and other structured proteins, we have developed an assay called 'Tite-Seq.' Tite-Seq, like SORTCERY, builds on the capabilities of Sort-Seq, an experimental strategy that was first developed for studying transcriptional regulatory sequences in bacteria (*Kinney et al., 2010*). Sort-Seq combines fluorescence-activated cell sorting (FACS) with high-throughput sequencing to provide massively parallel measurements of cellular fluorescence. In the Tite-Seq assay, Sort-Seq is applied to antibodies displayed on the surface of yeast cells and incubated with antigen at a wide range of concentrations. From the resulting sequence data, thousands of antibody-antigen binding titration curves and their corresponding absolute dissociation constants (here denoted $K_D$) can be inferred. By assaying full binding curves, Tite-Seq is able to measure affinities over many orders of magnitude (We note that *Kowalsky et al. (2015)* have described yeast display DMS experiments performed at multiple concentrations. These data, however, were not used to reconstruct titration curves or infer quantitative $K_D$ values). Moreover, the resulting affinity values provided by Tite-Seq are not confounded by the (rather substantial) effect that sequence variation can have on either (a) the amount of protein expressed on the surface of cells or (b) the specific activity of displayed proteins (i.e., the fraction of protein molecules that are functional).

We demonstrated Tite-Seq on a protein library derived from a well-studied single-chain variable fragment (scFv) antibody specific to the small molecule fluorescein (*Boder and Wittrup, 1997*; *Boder et al., 2000*). Mutations were restricted to CDR1H and CDR3H regions, which are known to play an important role in the antigen recognition of this scFv (*Boder et al., 2000*; *Midelfort et al., 2004*). The resulting affinity measurements were validated with binding curves for a handful of clones measured using standard low-throughput flow cytometry. Our Tite-Seq measurements reveal both expected and unexpected differences between the effects of mutations in CDR1H and CDR3H. These data also shed light on structural aspects of antigen recognition that are independent of effects on antibody stability.

## Results

### Overview of Tite-Seq

Our general strategy is illustrated in *Figure 1*. First, a library of variant antibodies is displayed on the surface of yeast cells (*Figure 1A*). The composition of this library is such that each cell displays a single antibody variant, and each variant is expressed on the surface of multiple cells. Cells are then incubated with the antigen of interest, bound antigen is fluorescently labeled, and fluorescence-

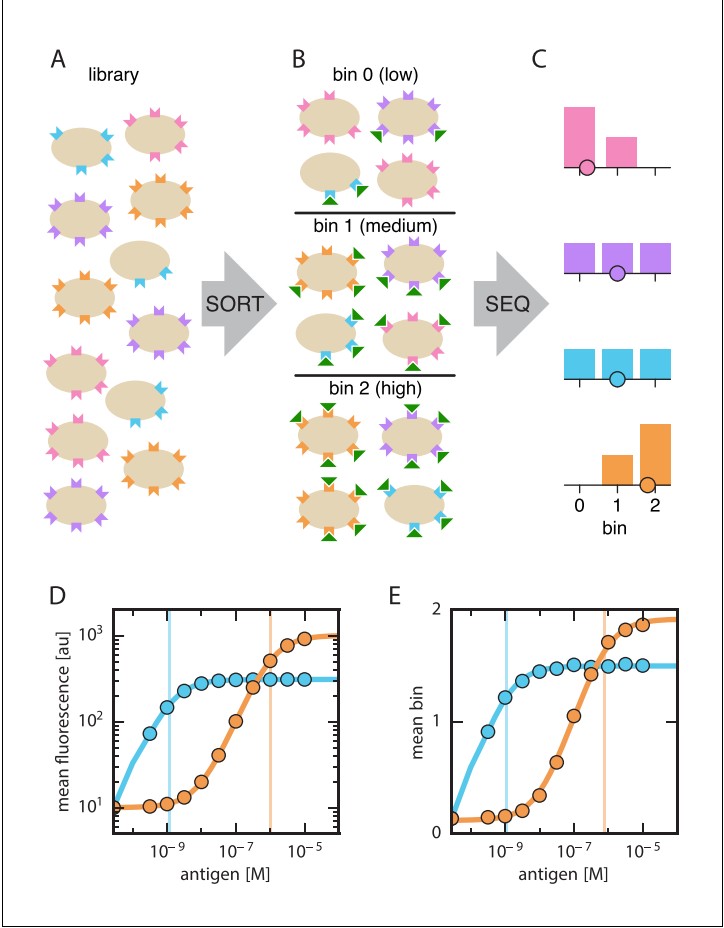

**Figure 1.** Schematic illustration of Tite-Seq. (**A**) A library of variant antibodies (various colors) are displayed on the surface of yeast cells (tan). (**B**) The library is exposed to antigen (green triangles) at a defined concentration, cell-bound antigen is fluorescently labeled, and FACS is used to sort cells into bins according to measured fluorescence. (**C**) The antibody variants in each bin are sequenced and the distribution of each variant across bins is computed (histograms; colors correspond to specific variants). The mean bin number (dot) is then used to quantify the typical amount of bound antigen per cell. (**D**) Binding titration curves (solid lines) and corresponding $K_D$ values (vertical lines) can be inferred for individual antibody sequences by using the mean fluorescence values (dots) obtained from flow cytometry experiments performed on clonal populations of antibody-displaying yeast. (**E**) Tite-Seq consists of performing the Sort–Seq experiment in panels **A–C** at multiple antigen concentrations, then inferring binding curves using mean bin number as a proxy for mean cellular fluorescence. This enables $K_D$ measurements for thousands of variant antibodies in parallel. We note that the Tite-Seq results illustrated in panel **E** were simulated using three bins under idealized experimental conditions, as described in Appendix 1. The inference of binding curves from real Tite-Seq data is more involved than this panel might suggest, due to the multiple sources of experimental noise that must be accounted for.

activated cell sorting (FACS) is used to sort cells one-by-one into multiple 'bins' based on this fluorescent readout (**Figure 1B**). Deep sequencing is then used to survey the antibody variants present in each bin. Because each variant antibody is sorted multiple times, it will be associated with a histogram of counts spread across one or more bins (**Figure 1C**). The spread in each histogram is due to cell-to-cell variability in antibody expression, and to the inherent noisiness of flow cytometry measurements. Finally, the histogram corresponding to each antibody variant is used to compute an 'average bin number' (**Figure 1C**, dots), which serves as a proxy measurement for the average amount of bound antigen per cell.

It has previously been shown that $K_D$ values can be accurately measured using yeast-displayed antibodies by taking binding titration curves, i.e., by measuring the average amount of bound

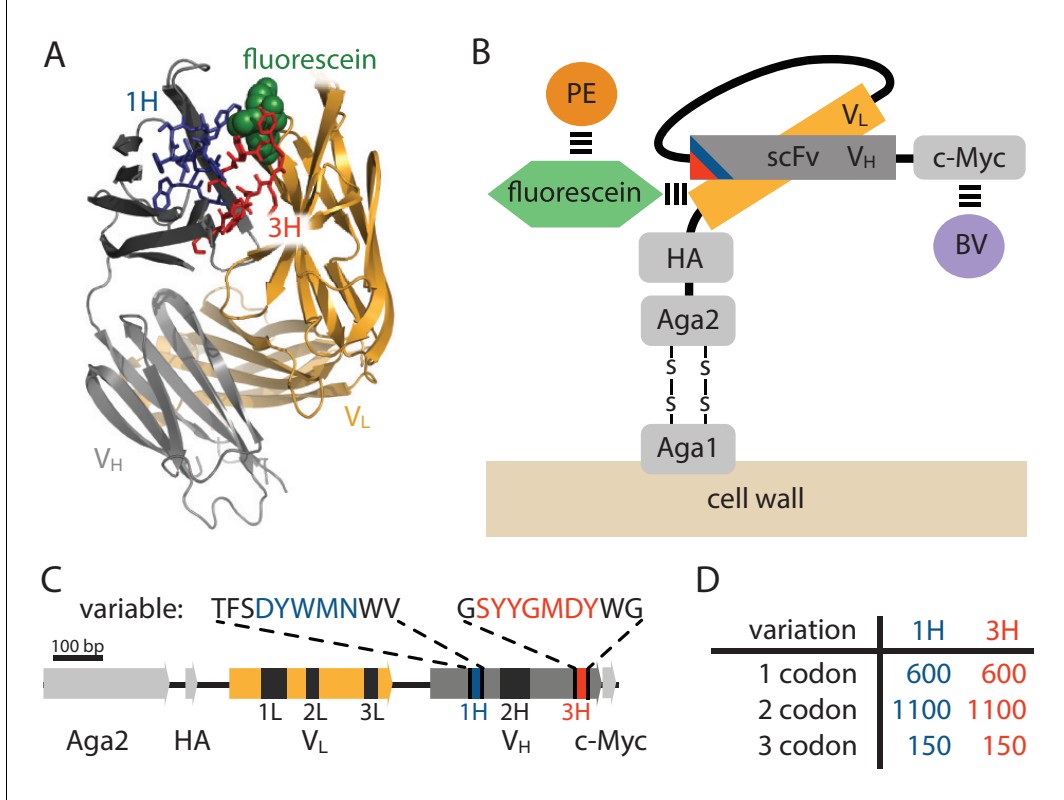

**Figure 2.** Yeast display construct and antibody libraries (**A**) Co-crystal structure of the 4-4-20 (WT) antibody from *Whitlow et al. (1995)* (PDB code 1FLR). The CDR1H and CDR3H regions are colored blue and red, respectively. (**B**) The yeast display scFv construct from *Boder and Wittrup (1997)* that was used in this study. Antibody-bound antigen (fluorescein) was visualized using PE dye. The amount of surface-expressed protein was separately visualized using BV dye. Approximate location of the CDR1H (blue) and CDR3H (red) regions within the scFv are illustrated. (**C**) The gene coding for this scFv construct, with the six CDR regions indicated. The WT sequence of the two 10 aa variable regions are also shown. (**D**) The number of 1-, 2-, and 3-codon variants present in the 1H and 3H scFv libraries. *Figure 2—figure supplement 1* shows the cloning vector used to construct the CDR1H and CDR3H libraries, as well as the form of the resulting expression plasmids.

The following figure supplement is available for figure 2:

**Figure supplement 1.** Cloning strategy.

antigen as a function of antigen concentration (*VanAntwerp and Wittrup, 2000*; *Gai and Wittrup, 2007*). The median fluorescence $f$ of labeled cells is expected to be related to antigen concentration via

$$f = A\frac{c}{c + K_D} + B \tag{1}$$

where $A$ is proportional to the number of functional antibodies displayed on the cell surface, $B$ accounts for background fluorescence, and $c$ is the concentration of free antigen in solution. *Figure 1D* illustrates the shape of curves having this form. By using flow cytometry to measure $f$ on clonal populations of yeast at different antigen concentrations $c$, one can infer curves having the sigmoidal form shown in *Equation 1* and thereby learn $K_D$. Such measurements, however, can only be performed in a low-throughput manner.

Tite-Seq allows thousands of binding titration curves to be measured in parallel. The Sort-Seq procedure illustrated in *Figure 1A–C* is performed at multiple antigen concentrations, and the

resulting average bin number for each variant antibody is plotted against concentration. Sigmoidal curves are then fit to these proxy measurements, enabling $K_D$ values to be inferred for each variant.

We emphasize that $K_D$ values cannot, in general, be accurately inferred from Sort-Seq experiments performed at a single antigen concentration. Because the relationship between binding and $K_D$ is sigmoidal, the amount of bound antigen provides a quantitative readout of $K_D$ only when the concentration of antigen used in the labeling procedure is comparable in magnitude to $K_D$. However, single mutations within a protein binding domain often change $K_D$ by multiple orders of magnitude. Sort-Seq experiments used to measure sequence-affinity landscapes must therefore be carried out over a range of concentrations large enough to encompass this variation.

Furthermore, as illustrated in *Figure 1C and D*, different antibody variants often lead to different levels of functional antibody expression on the yeast cell surface. If one performs Sort-Seq at a single antigen concentration, high affinity (low $K_D$) variants with low expression (blue variant) may bind less antigen than low affinity (high $K_D$) variants with high expression (orange variant). Only by measuring full titration curves can the effect that sequence has on affinity be deconvolved from sequence-dependent effects on functional protein expression.

## Proof-of-principle Tite-Seq experiments

To test the feasibility of Tite-Seq, we used a well-characterized antibody-antigen system: the 4-4-20 single chain variable fragment (scFv) antibody (*Boder and Wittrup, 1997*), which binds the small molecule fluorescein with $K_D = 1.2$ nM (*Gai and Wittrup, 2007*). This system was used in early work to establish the capabilities of yeast display (*Boder and Wittrup, 1997*), and a high resolution co-crystal structure of the 4-4-20 antibody bound to fluorescein, shown in *Figure 2A*, has been determined (*Whitlow et al., 1995*). An ultra-high-affinity ($K_D = 270$ fM) variant of this scFv, called 4m5.3, has also been found (*Boder et al., 2000*). In what follows, we refer to the 4-4-20 scFv from *Boder and Wittrup (1997)* as WT, and the 4m5.3 variant from *Boder et al. (2000)* as OPT.

The scFv was expressed on the surface of yeast as part of the multi-domain construct illustrated in *Figure 2B* and previously described in *Boder and Wittrup (1997)*. Following (*Boder et al., 2000*), we used fluorescein-biotin as the antigen and labeled scFv-bound antigen with streptavidin-RPE (PE). The amount of surface-expressed protein was separately quantified by labeling the C-terminal c-Myc tag using anti-c-Myc primary antibodies, followed by secondary antibodies conjugated to Brilliant Violet 421 (BV). See Appendix 2 for details on this labeling procedure.

Two different scFv libraries were assayed simultaneously. In the '1H' library, a 10 aa region encompassing the CDR1H region of the WT scFv (see *Figure 2C*) was mutagenized using microarray-synthesized oligos (see Appendix 3 for details). The resulting 1H library consisted of all 600 single-codon variants of this 10 aa region, 1100 randomly chosen 2-codon variants, and 150 random 3-codon variants (*Figure 2D*). An analogous '3H' library was generated for a 10 aa region containing the CDR3H region of this scFv. In all of the Tite-Seq experiments described below, these two libraries were pooled together and supplemented with WT and OPT scFvs, as well with a nonfunctional scFv referred to as Δ.

Tite-Seq was carried out as follows. Yeast cells expressing scFv from the mixed library were incubated with fluorescein-biotin at one of eleven concentrations: 0 M, $10^{-9.5}$ M, $10^{-9}$ M, $10^{-8.5}$ M, $10^{-8}$ M, $10^{-7.5}$ M, $10^{-7}$ M, $10^{-6.5}$ M, $10^{-6}$ M, $10^{-5.5}$ M, and $10^{-5}$ M. After subsequent PE labeling of bound antigen, cells were sorted into four bins using FACS (*Figure 3A*). Separately, BV-labeled cells were sorted according to measured scFv expression levels (*Figure 3B*). The number of cells sorted into each bin is shown in *Figure 3C*. Each bin of cells was regrown and bulk DNA was extracted. The 1H and 3H variable regions were then PCR amplified and sequenced using paired-end Illumina sequencing, as described in Appendix 4. The final data set consisted of an average of $2.6 \times 10^6$ sequences per bin across all 48 bins (*Figure 3D*). Three independent replicates of this experiment were performed on three different days.

For each variant scFv gene, a $K_D$ value was inferred by fitting a binding curve to the resulting Tite-Seq data, with separate curves independently fit to data from each Tite-Seq experiment (*Figure 4A*). As illustrated in *Figure 1E*, this fitting procedure uses the sigmoidal function in *Equation 1* to model mean bin number as a function of antigen concentration. However, the need to account for multiple sources of noise in the Tite-Seq experiment necessitates a more complex

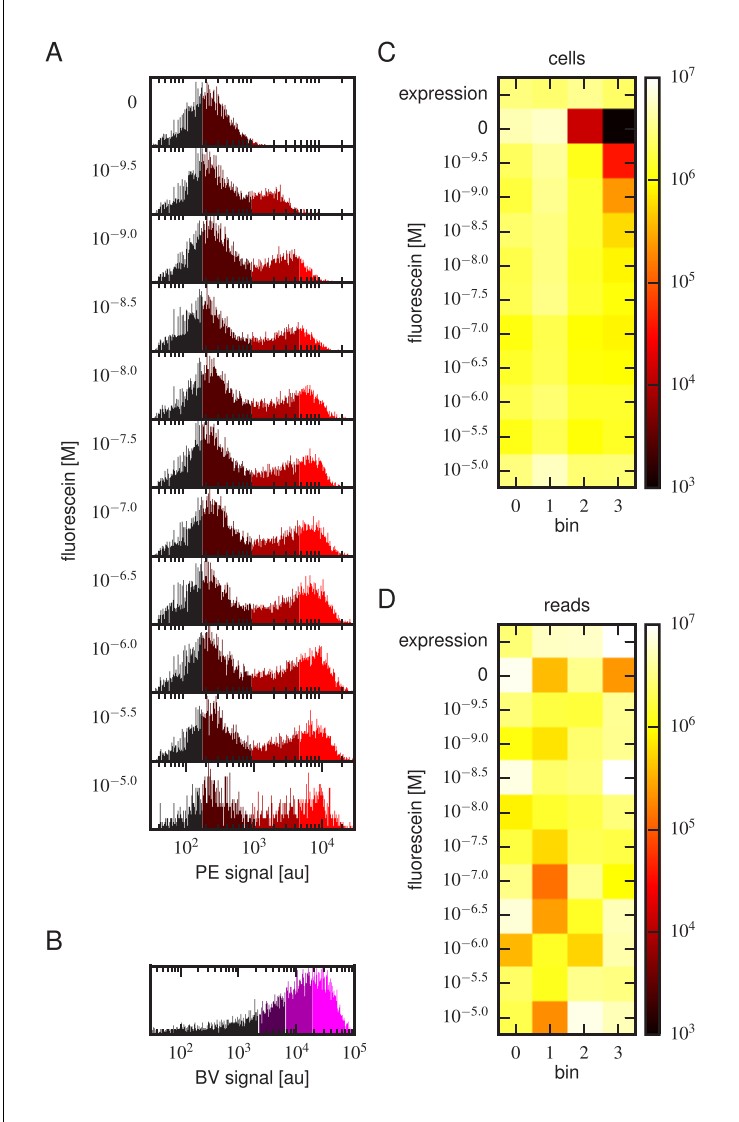

**Figure 3.** Details of our Tite-Seq experiments. (**A**) Gates used to sort cells based on PE fluorescence, which provides a readout of bound antigen. Cells were labeled at the eleven different antigen concentrations. Shades of red indicate the four fluorescence gates used to sort cells; these correspond to bins 0, 1, 2, and 3 (from left to right). (**B**) Gates, indicated in shades of purple, used to sort cells based on BV fluorescence, which provides a readout of antibody expression. (**C**) The number of cells sorted into each bin. (**D**) The number of Illumina reads obtained from each bin of sorted cells after quality control measures were applied. The data shown in this figure corresponds to a single Tite-Seq experiment. *Figure 3—figure supplement 1* and *Figure 3—figure supplement 2* show data for two independent replicates of this experiment.

The following figure supplements are available for figure 3:

**Figure supplement 1.** Tite-Seq experiment, replicate 2.

**Figure supplement 2.** Tite-Seq experiment, replicate 3.

procedure than *Figure 1E* might suggest; the details of this inference procedure are described in Appendix 5.

Separately, the Sort-Seq data obtained by sorting the BV-labeled libraries were used to determine the expression level of each scFv. Specifically, we use $E$ to denote (for each scFv in the library)

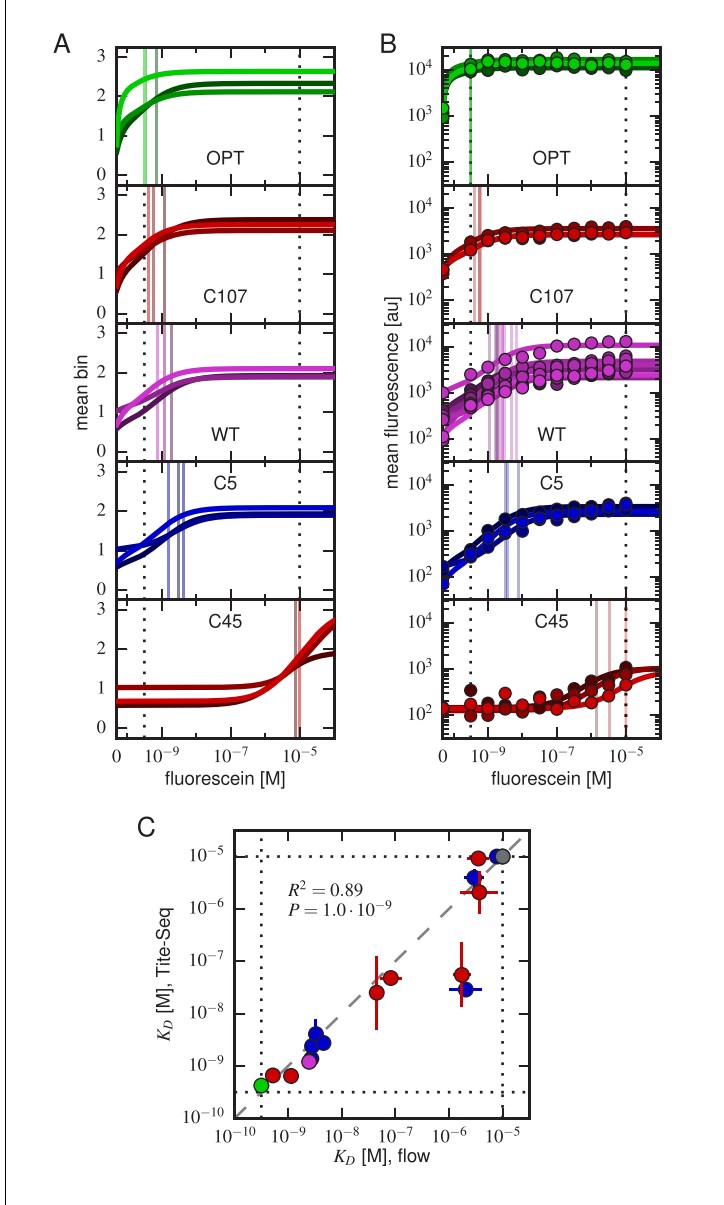

**Figure 4.** Accuracy and precision of Tite-Seq. (**A**) Binding curves and $K_D$ measurements inferred from Tite-Seq data. (**B**) Mean fluorescence values (dots) and corresponding inferred binding curves (lines) obtained by flow cytometry measurements for five selected scFvs (WT, OPT, C5, C45, and C107). In (**A,B**), values corresponding to 0 M fluorescein are plotted on the left-most edge of the plot, dotted lines show the upper ($10^{-5}$ M) and lower ($10^{-9.5}$ M) limits on $K_D$ sensitivity, vertical lines show inferred $K_D$ values, and different shades correspond to different replicate experiments. (**C**) Comparison of the Tite-Seq-measured and flow-cytometry-measured $K_D$ values for all clones tested. Colors indicate different scFv protein sequences as follows: WT (purple), OPT (green), Δ (black), 1H clones (blue), and 3H clones (red). Each $K_D$ value indicates the mean $\log_{10} K_D$ value obtained across all replicates, with error bars indicating standard error. Clones with $K_D$ outside of the affinity range are drawn on the boundaries of this range, which are indicated with dotted lines. The coefficient of determination ($R^2$) between log Tite-Seq values and log flow $K_D$ values includes clones outside of the affinity range; in such cases, the corresponding boundary value ($10^{-9.5}$ M or $10^{-5.0}$ M) has been used. The amino acid sequences and measured $K_D$ values for all clones tested are provided in *Table 1*. *Figure 4—figure supplement 1* provides plots, analogous to panels **A** and **B**, for all of the assayed clones. *Figure 4—figure supplement 2* compares $K_D$ and $E$ values obtained across all three Tite-Seq replicates. *Figure 4—figure supplement 3* quantifies measurement error using synonymous mutants. *Figure 4—figure supplement 4* provides information about library composition. *Figure 4—figure supplement 5* illustrates the poor correlation between scFv enrichment and Tite-seq measured $K_D$ values.

*Figure 4 continued on next page*

*Figure 4 continued*

*Figure 4—figure supplement 6* shows a 2-fold difference in the specific activities of OPT and WT scFvs.
*Figure 4—figure supplement 7* illustrates the simulations we used in *Figure 4—figure supplement 8* to validate the ability of our analysis to infer correct $K_D$ values.

The following figure supplements are available for figure 4:

**Figure supplement 1.** Binding curves for all clones.
**Figure supplement 2.** Concordance between replicate experiments.
**Figure supplement 3.** Error estimates from synonymous mutants.
**Figure supplement 4.** Composition of scFv libraries.
**Figure supplement 5.** Sort-Seq enrichment correlates poorly with Tite-Seq-measured affinity.
**Figure supplement 6.** Differing specific activities of OPT and WT.
**Figure supplement 7.** Realistic Tite-Seq simulations.
**Figure supplement 8.** Validation of analysis pipeline.

the mean bin number that results from this expression-based sorting; this $E$ value provides a measurement of the surface expression level of that scFv. All $E$ values have been scaled so that the mean of such measurements for all synonymous WT scFv gene variants is 1.0.

## Low-throughput validation experiments

To judge the accuracy of Tite-Seq, we separately measured binding curves for individual scFv clones as described for *Figure 1D*. In addition to the WT, OPT, and Δ scFvs, we assayed eight clones from the 1H library (named C3, C5, C7, C18, C22, C132, C133 and C144) and eight clones from the 3H library (C39, C45, C93, C94, C102, C103, C107, C112). Each clone underwent the same labeling procedure as in the Tite-Seq experiment, after which median fluorescence values were measured using standard flow cytometry. $K_D$ values were then inferred by fitting binding curves of the form in *Equation 1* using the procedure described in Appendix 6. These curves, which can be directly compared to Tite-Seq measurements (*Figure 4A*), are plotted in *Figure 4B*; at least three replicate binding curves were measured for each clone. See *Figure 4—figure supplement 1* for the titration curves of all the tested clones.

## Tite-Seq can measure dissociation constants

*Figure 4C* reveals a strong correspondence between the $K_D$ values measured by Tite-Seq and those measured using low-throughput flow cytometry. The robustness of Tite-Seq is further illustrated by the consistency of $K_D$ values measured for the WT scFv. Using Tite-Seq, and averaging the results from the 33 synonymous variants and over all three replicates, we determined $K_D = 10^{-8.87 \pm 0.02}$ M for the WT scFv. These measurements are largely consistent with the measurement of $K_D = 10^{-8.61 \pm 0.07}$ M obtained by averaging low-throughput flow cytometry measurements across 10 replicates, and coincides with the previously measured value of $1.2$ nM $= 10^{-8.9}$ M reported in (*Gai and Wittrup, 2007*). The three independent replicate Tite-Seq experiments give reproducible results as measured by direct comparison (*Figure 4—figure supplement 2*), from synonymous mutant variation (*Figure 4—figure supplement 3*) and library composition *Figure 4—figure supplement 4*) with Pearson coefficients ranging from $r = 0.82$ to $r = 0.89$ for all the measured $K_D$ values between replicates; note that $K_D$ values outside of the sensitivity range are included in the calculation of these Pearson coefficients as described in the *Figure 4* caption.

The error bars for $K_D$ values in *Figure 4C* calculated from the variability of the fits to different replicates therefore support the reproducibility of the experiment. The main discrepancy in these error

**Table 1.** Clones measured using flow cytometry and Tite-Seq. List of scFv clones, ordered by their flow-cytometry-measured $K_D$ values. With the exception of OPT and Δ, these clones differed from WT only in their 1H and 3H variable regions. WT amino acids within these regions are capitalized; variant amino acids are shown in lower case. No sequence is shown for Δ because this clone contained a large deletion, making identification of the 1H and 3H variable regions meaningless. $K_D$ values saturating our lower detection limit of $10^{-9.5}$ M or upper detection limit of $10^{-5.0}$M are written with a ≤ or ≥ sign to emphasize the uncertainty in these measurements. Tite-Seq $K_D$ values indicate mean and standard errors computed across the three replicate Tite-Seq experiments; they are not averaged across synonymous variants.

| Name | 1H variable region | 3H variable region | No. replicates (flow) | $K_D$ [M] (flow) | $K_D$ [M] (Tite-Seq) |
|---|---|---|---|---|---|
| OPT | TFghYWMNWV | GasYGMeYlG | 3 | $\leq 10^{-9.5}$ | $\leq 10^{-9.5}$ |
| C107 | TFSDYWMNWV | GaYYGMDYWG | 3 | $10^{-9.28\pm0.04}$ | $10^{-9.18\pm0.11}$ |
| C112 | TFSDYWMNWV | GSYYGMDYcG | 3 | $10^{-8.95\pm0.07}$ | $10^{-9.19\pm0.14}$ |
| WT | TFSDYWMNWV | GSYYGMDYWG | 10 | $10^{-8.61\pm0.07}$ | $10^{-8.92\pm0.10}$ |
| C144 | vFSDYWMNWV | GSYYGMDYWG | 3 | $10^{-8.57\pm0.03}$ | $10^{-8.86\pm0.04}$ |
| C133 | aFSDYWMNWV | GSYYGMDYWG | 3 | $10^{-8.55\pm0.06}$ | $10^{-8.62\pm0.09}$ |
| C132 | TFmDYWlNWV | GSYYGMDYWG | 3 | $10^{-8.48\pm0.08}$ | $10^{-8.38\pm0.29}$ |
| C94 | TFSDYWMNWV | GSYYGMDsWG | 3 | $10^{-8.46\pm0.06}$ | $10^{-8.50\pm0.04}$ |
| C5 | TFSDYWiNWV | GSYYGMDYWG | 3 | $10^{-8.34\pm0.10}$ | $10^{-8.55\pm0.09}$ |
| C93 | TFSDYWMNWV | GSYrGMDYWG | 3 | $10^{-7.35\pm0.08}$ | $10^{-7.60\pm0.70}$ |
| C39 | TFSDYWMNWV | GSYYGMDYWa | 3 | $10^{-7.08\pm0.20}$ | $10^{-7.28\pm0.17}$ |
| C102 | TFSDYWMNWV | sSkYGMDYWG | 3 | $10^{-5.76\pm0.16}$ | $10^{-7.25\pm0.60}$ |
| C22 | ssSDYWMNWV | GSYYGMDYWG | 3 | $10^{-5.69\pm0.31}$ | $10^{-7.53\pm0.07}$ |
| C7 | hFSDYWMNWl | GSYYGMDYWG | 3 | $10^{-5.53\pm0.18}$ | $10^{-5.39\pm0.18}$ |
| C45 | TFSDYWMNWV | GSYdGnDYWG | 3 | $10^{-5.40\pm0.24}$ | $\geq 10^{-5.0}$ |
| C103 | TFSDYWMNWV | GSYYGMDlWG | 3 | $10^{-5.15\pm0.47}$ | $10^{-5.44\pm0.55}$ |
| C3 | TFSDYWMsWV | GSYYGMDYWG | 3 | $\geq 10^{-5.0}$ | $\geq 10^{-5.0}$ |
| C18 | TFSDYsMNWV | GSYYGMDYWG | 3 | $\geq 10^{-5.0}$ | $\geq 10^{-5.0}$ |
| Δ | – | – | 12 | $\geq 10^{-5.0}$ | $\geq 10^{-5.0}$ |

bar calculations occurred for clones c22 and c102 (see also *Figure 4—figure supplement 1*). The reason for this discrepancy is currently unclear. We note that Tite-Seq-measured $K_D$ values for these two clones are close to $10^{-7}$ M, and that the analysis of synonymous variants (*Figure 4—figure supplement 3*) found that Tite-Seq-measured $K_D$s in this region exhibited the largest variations.

The necessity of performing $K_D$ measurements over a wide range of antigen concentrations is illustrated in *Figure 4—figure supplement 5*. At each antigen concentration used in our Tite-Seq experiments, the enrichment of scFvs in the high-PE bins correlated poorly with the $K_D$ values inferred from full titration curves. Moreover, at each antigen concentration used, a detectable correlation between $K_D$ and enrichment was found only for scFvs with $K_D$ values close to that concentration.

*Figure 4—figure supplement 6* suggests a possible reason for the weak correlation between $K_D$ values and enrichment in high-PE bins. We found that, at saturating concentrations of fluorescein (2 μM), cells expressing the OPT scFv bound twice as much fluorescein as cells expressing the WT scFv. This difference was not due to variation in the total amount of displayed scFv, which one might control for by labeling the c-Myc epitope as in *Reich et al. (2015)*. Rather, this difference in binding reflects a difference in the specific activity of displayed scFvs. Yeast display experiments performed at a single antigen concentration cannot distinguish such differences in specific activity from differences in scFv affinity.

To further test the capability of Tite-Seq to infer dissociation constants from sequencing data over a wide range of values, as well as to validate our analysis procedures, we simulated Tite-Seq

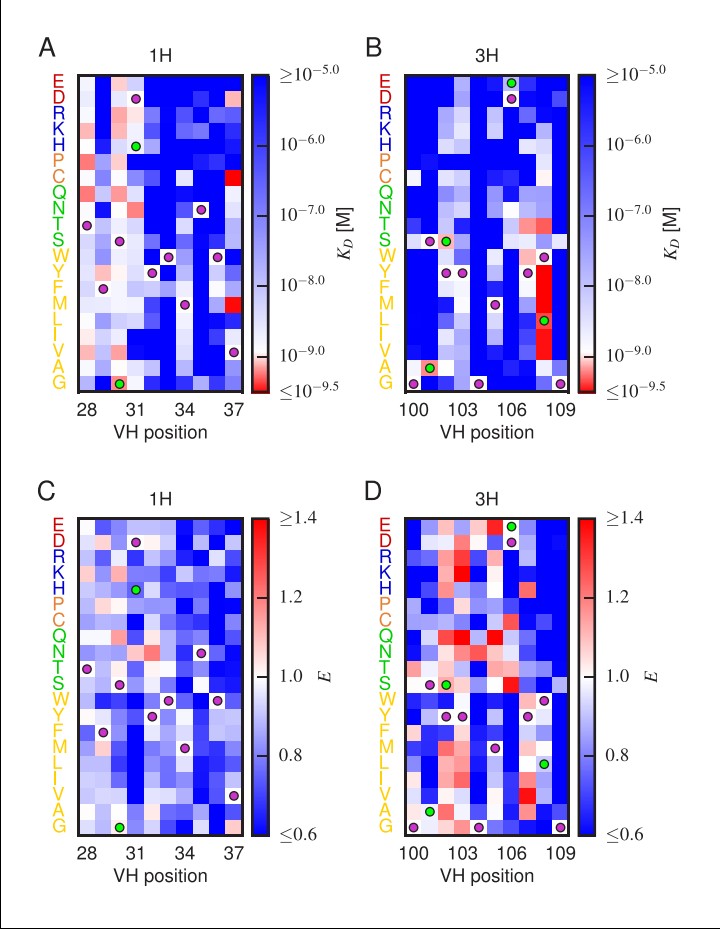

**Figure 5.** Effects of substitution mutations on affinity and expression. Heatmaps show the measured effects on affinity (**A,B**) and expression (**C,D**) of all single amino acid substitutions within the variables regions of the 1H (**A,C**) and 3H (**B,D**) libraries. Purple dots indicate residues of the WT scFv. Green dots indicate non-WT residues in the OPT scFv. *Figure 5—figure supplement 1* provides histograms of the non-WT values displayed in panels **A–D**. *Figure 5—figure supplement 2* compares the effects on $K_D$ of both single-point and multi-point mutations.

The following figure supplements are available for figure 5:

**Figure supplement 1.** Histograms of substitution effects on affinity and expression.

**Figure supplement 2.** Effects of multi-point mutations on affinity and expression.

data in silico and analyzed the results using the same analysis pipeline that we used for our experiments. Details about the simulations are given in Appendix 7. The simulated data is illustrated in *Figure 4—figure supplement 7*. $K_D$ values inferred from these simulated data agreed to high accuracy with the $K_D$ used in the simulation (*Figure 4—figure supplement 8*), thus validating our analysis pipeline.

## Properties of the affinity and expression landscapes

*Figure 5* shows the effect that every single-amino-acid substitution mutation within the 1H and 3H variable regions has on affinity and on expression; histograms of these effects are provided in *Figure 5—figure supplement 1*. In both regions, the large majority of mutations weaken antigen binding (1H: 88%; 3H: 93%), with many mutations increasing $K_D$ above our detection threshold of $10^{-5}$ M (1H: 36%; 3H: 52%). Far fewer mutations reduced $K_D$ (1H: 12%; 3H: 7%), and very few dropped $K_D$ below our detection limit of $10^{-9.5}$ M (1H: 0%; 3H: 3%). Histograms of the effect of two or three

amino acid changes relative to WT, shown in *Figure 5—figure supplement 2A*, reveal that multiple random mutations tend to further reduce affinity. We also observed that mutations within the 3H variable region have a larger effect on affinity than do mutations in the 1H variable region. Specifically, single amino acid mutations in 3H were seen to increased $K_D$ more than mutations in 1H (1H median $K_D = 10^{-6.84}$; 3H median $K_D \geq 10^{-5.0}$; $P = 4.7 \times 10^{-4}$, one-sided Mann-Whitney U test). This result suggests that binding affinity is more sensitive to variation in CDR3H than to variation in CDR1H, a finding that is consistent with the conventional understanding of these antibody CDR regions (*Xu and Davis, 2000*; *Liberman et al., 2013*).

Our observations are thus fully consistent with the hypothesis that the amino acid sequences of the CDR1H and CDR3H regions of the WT scFv have been selected for high affinity binding to fluorescein. We know this to be true, of course; still, this result provides an important validation of our Tite-Seq measurements.

To further validate our Tite-Seq affinity measurements, we examined positions in the high affinity OPT scFv (from [*Boder et al., 2000*]) that differ from WT and that lie within the 1H and 3H variable regions. As illustrated in *Figure 5A and B*, five of the six OPT-specific mutations reduce $K_D$ or are nearly neutral. Previous structural analysis (*Midelfort et al., 2004*) has suggested that D106E, the only OPT mutation that we find significantly increases $K_D$, may indeed disrupt antigen binding on its own while still increasing affinity in the presence of the S101A mutation.

Next, we used our measurements to build a 'matrix model' (also known as a 'position-specific affinity matrix,' or PSAM [*Foat et al., 2006*]) describing the sequence-affinity landscape of these two regions. Our model assumed that the $\log_{10} K_D$ value for an arbitrary amino acid sequence could be computed from the $\log_{10} K_D$ value of the WT scFv, plus the measured change in $\log_{10} K_D$ produced by each amino acid substitution away from WT. We evaluated our matrix models on the 1H and 3H variable regions of OPT, finding an affinity of $10^{-9.16}$ M. Our simple model for the sequence affinity landscape of this scFv therefore correctly predicts that OPT has higher affinity than WT. The quantitative affinity predicted by our model does not match the known affinity of the OPT scFv ($K_D = 10^{-12.6}$ M), but this is unsurprising for three reasons. First the OPT scFv differs from WT in 14 residues, only 6 of which are inside the 1H and 3H variable regions assayed here. Second, one of the OPT mutations (W108L) reduces $K_D$ below our detection threshold of $10^{-9.5}$ M; in building our matrix model, we set this value equal to $10^{-9.5}$, knowing it would likely underestimate the affinity-increasing effect of the mutation. Third, our additive model ignores potential epistatic interactions. Still, we thought it worth asking how likely it it would be for six random mutations within the 1H and 3H variable regions to reduce affinity as much as our model predicts for OPT. We therefore simulated a large number ($10^7$) of variants having a total of 6 substitution mutations randomly scattered across the 1H and 3H variable regions. The fraction of these random sequences that had an affinity at or below our predicted affinity for OPT was $4.7 \times 10^{-5}$. This finding is fully consistent with the fact that the mutations in OPT relative to WT were selected for increased affinity, an additional confirmation of the validity of our Tite-Seq measurements.

The sequence-expression landscape measured in our separate Sort-Seq experiment yielded qualitatively different results (*Figure 5C and D*). We observed no significant difference in the median effect that mutations in the variable regions of 1H (median $E = 0.826$) versus 3H (median $E = 0.822$) have on expression ($P = 0.96$, two-sided Mann-Whitney U test); see also *Figure 5—figure supplement 1*. The variance in these effects, however, was larger in 3H than in 1H ($P = 9.9 \times 10^{-16}$, Levene's test). These results suggest two things. First, the 3H variable region appears to have a larger effect on scFv expression than the 1H variable region has. At the same time, since we observe fewer beneficial mutations in 1H (*Figure 5C*) than in 3H (*Figure 5D*), the WT sequence appears to be more highly optimized for expression in CDR1H than in CDR3H. The effect of double or triple mutations further reduced expression in both CDRs (*Figure 5—figure supplement 2B*), similar to what was observed for affinity.

## Structural correlates of the sequence-affinity landscape

We asked if the sensitivity of the antibody to mutations could be understood from a structural perspective. To quantify sensitivity of affinity and expression at each position $i$, we computed two quantities:

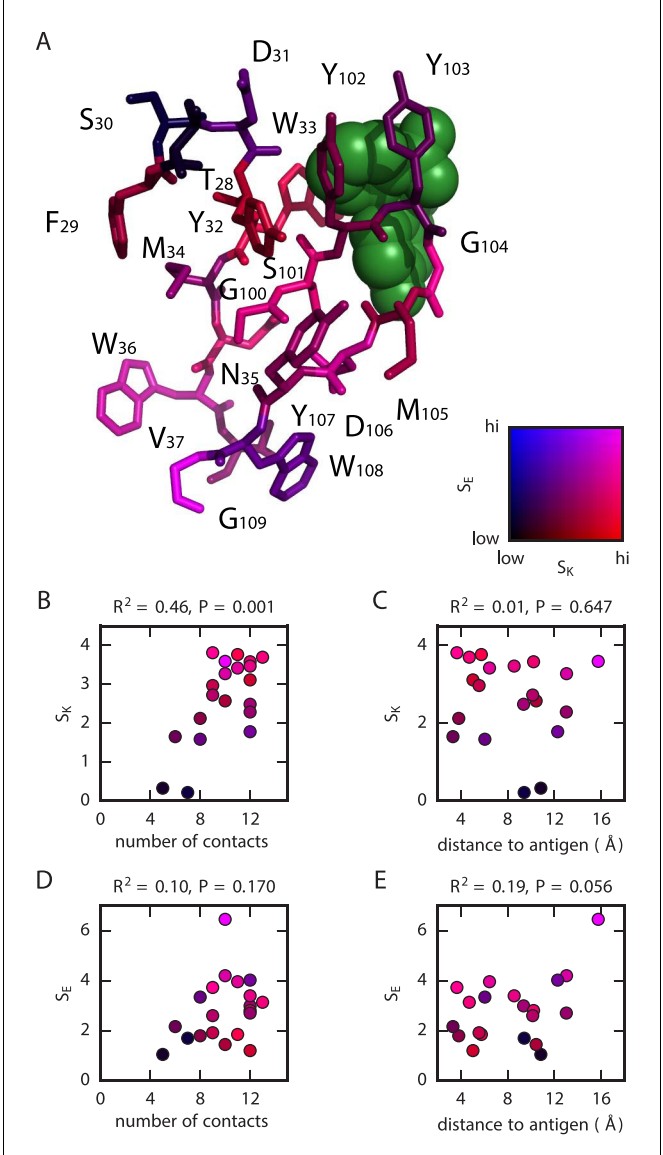

**Figure 6.** Structural context of mutational effects. (**A**) Crystal structure (*Whitlow et al., 1995*) of the CDR1H and CDR3H variable regions of the WT scFv in complex with fluorescein (green). Each residue (CDR1H: positions 28–37; CDR3H: positions 100–109) is colored according to the $S_K$ and $S_E$ values computed for that position. These variables, $S_K$ and $S_E$, respectively quantify the sensitivity of $K_D$ and $E$ to amino acid substitutions at each position, with larger values corresponding to greater sensitivity; see *Equations 2 and 3* for definitions of these quantities. (**B,C**) For each position in the CDR1H and CDR3H variable regions, $S_K$ is plotted against either (**B**) the number of contacts the WT residue makes within the protein structure, or (**C**) the distance of the WT residue to the fluorescein molecule. (**D,E**) Similarly, $S_E$ is plotted against either (**D**) the number of contacts or (**E**) the distance to the antigen. $R^2$ is the coefficient of determination.

$$S_K^i = \sqrt{\left\langle \left(\log_{10} K_D^{ia} - \log_{10} K_D^{\mathrm{WT}}\right)^2 \right\rangle_{a|i}},$$ (2)

$$S_E^i = \sqrt{\left\langle \left(E^{ia} - E^{\mathrm{WT}}\right)^2 \right\rangle_{a|i}}.$$ (3)

Here, $K_D^{\mathrm{WT}}$ and $E^{\mathrm{WT}}$ respectively denote the dissociation constant and expression level measured

**Table 2.** Primers. Oligonucleotide sequences are written 5′ to 3′. Bold sequences indicate variable regions. The '1H library' and '3H library' primers respectively contained the 1H and 3H variable regions (bold) analyzed in this paper. These primer libraries were synthesized by LC Biosciences using microarray-based DNA synthesis. All other primers were ordered from Integrated DNA Technologies. The '[XX]' portion of L1AF_XX and L1AR_XX indicates the location of each of 64 different barcodes (i.e., XX = 01, 02, . . ., 64), which ranged in length from 7 bp to 10 bp and which differed from each other by at least two substitution mutations.

| Name | Sequence |
|---|---|
| 1H library | GTGTTGCCTCTGGATTC**ACTTTTAGTGACTACTGGATGAACTGGGTC**CGCCAGTCTCCAGA |
| 3H library | GTGACTGAGGTTCCTTG**ACCCCAGTAGTCCATACCATAGTAAGAACC**CGTACAGTAATAGATACCCAT |
| oRAL10 | TTCTGAGGAGACGGTGACTGAGGTTCCTTG |
| oRAR10 | TGAAGACATGGGTATCTATTACTGTACG |
| oRAL11 | CAGTCCTTTCTCTGGAGACTGGCG |
| oRAR11 | ATGAAACTCTCCTGTGTTGCCTCTGGATTC |
| 3H1F | TTCTGAGGAGACGGTGACT |
| 3H2R | TGAAGACATGGGTATCTATTACTGTAC |
| 1H2F | CAGTCCTTTCTCTGGAGACTG |
| 1H1R | ATGAAACTCTCCTGTGTTGCCT |
| oRA10 | GCATATCTAAGGTCTCGTTCTGAGGAGACGGTGAC |
| oRA11 | GCCGATTGTTGGTCTCCATGAAACTCTCCTGTGTTGC |
| PE1v3ext | AATGATACGGCGACCACCGAGATCTACACTCTTTCCCTACACGACG |
| PE2v3 | AAGCAGAAGACGGCATACGAGATCGGTCTCGGCATTCCTGCT |
| L1AF_XX | ACACTCTTTCCCTACACGACGCTCTTCCGATCT**[XX]**AGTCTTCTTCAGAAATAAGC |
| L1AR_XX | CTCGGCATTCCTGCTGAACCGCTCTTCCGATCT**[XX]**GCTTGGTGCAACCTG |

for the WT scFv, $K_D^{ia}$ and $E^{ia}$ denote analogous quantities for the scFv with a single substitution mutation of amino acid $a$ at position $i$, and $\langle \cdot \rangle_{a|i}$ denotes an average computed over the 19 non-WT amino acids at that position.

*Figure 6A* shows the known structure (*Whitlow et al., 1995*) of the 1H and 3H variable regions of the WT scFv in complex with fluorescein. Each residue is colored according to the $S_K$ and $S_E$ values computed for its position. To get a better understanding of what aspects of the structure might govern affinity, we plotted $S_K$ values against two other quantities: the number of amino acid contacts made by the WT residue within the antibody structure (*Figure 6B*), and the distance between the WT residue and the antigen (*Figure 6C*). We found a strong correlation between $S_K$ and the number of contacts, but no significant correlation between $S_K$ and distance to antigen. By contrast, $S_E$ did not correlate significantly with either of these structural quantities (*Figure 6D and E*).

## Discussion

We have described a massively parallel assay, called Tite-Seq, for measuring the sequence-affinity landscape of antibodies. The range of affinities measured in our Tite-Seq experiments ($10^{-9.5}$ M to $10^{-5.0}$ M) includes a large fraction of the physiological range relevant to affinity maturation ($10^{-10}$ M to ~$10^{-6}$ M) (*Batista and Neuberger, 1998*; *Foote and Eisen, 1995*; *Roost et al., 1995*). Expanding the measured range of affinities below $10^{-9.5}$ M might require larger volume labeling reactions, but would be straight-forward. Tite-Seq therefore provides a potentially powerful method for mapping the sequence-affinity trajectories of antibodies during the affinity maturation process, as well as for studying other aspects of the adaptive immune response.

The details of our Tite-Seq experiments (e.g., 11 antigen concentrations, four sorting bins per concentration, etc.) were chosen largely for experimental convenience. The effects of varying these parameters have not been systematically explored, and a future investigation of these effects might be valuable. *Figure 4—figure supplement 8* does illustrate, via simulation, the effect of read depth on the precision of measured $K_D$ values. These simulations, along with an analysis of synonymous

variants (*Figure 4—figure supplement 3*), suggest that the primary source of noise in our experiments came not from a lack of sorted cells or Illumina reads, but rather from the inefficient post-sort recovery of antibody sequences. We therefore suggest that improvements to our post-sort DNA recovery protocol might substantially improve the resolution of Tite-Seq.

Tite-Seq fundamentally differs from prior DMS experiments in that full binding titration curves, not two-bin enrichment statistics, are used to determine binding affinities. The measurement of binding curves provides three major advantages. First, binding curves provide absolute $K_D$ values in molar units, not just rank-order affinities, like those provided by SORTCERY (*Reich et al., 2015*), or relative affinity ratios, like those provided by the method of *Kowalsky and Whitehead (2016)*. Second, because ligand binding is a sigmoidal function of affinity, DMS experiments performed at a single ligand concentration (e.g., [*Kowalsky and Whitehead, 2016*]) are insensitive to receptor $K_D$s that differ substantially from this ligand concentration. Binding curves, by contrast, integrate measurements over a wide range of concentrations and are therefore sensitive to a wide range of $K_D$s.

The third advantage of measuring binding curves pertains to the fact that protein sequence determines not just ligand-binding affinity, but also the quantity and specific activity of surface-displayed proteins. Our data (*Figure 4—figure supplement 5* and *Figure 4—figure supplement 6*) suggest that these confounding effects can be large and that they can distort yeast display affinity measurements computed from enrichment statistics gathered at a single antigen concentration. Strong sequence-dependent effects on both the expression and specific activity of yeast-displayed proteins has been reported by other groups as well (e.g., [*Burns et al., 2014*]), although the absence of such effects has also been reported (e.g., [*Kowalsky and Whitehead, 2016*]). Ultimately, the magnitude of these effects is likely to vary substantially from protein to protein. It should also be noted that many DMS studies using yeast display (e.g., epitope mapping studies [*Kowalsky et al., 2015*; *Doolan and Colby, 2015*; *Van Blarcom et al., 2015*]) might not suffer from these potentially confounding effects, and in such cases it probably makes sense to employ a simpler experimental design than is required for Tite-Seq. Nevertheless, either Tite-Seq or other experimental methods that assay full binding curves are probably essential if one wants to quantitatively and reliably measure $K_D$ values in a massively parallel fashion.

We wish to emphasize, more generally, that changing a protein's amino acid sequence can be expected to change multiple biochemical properties of that protein. Our work illustrates the importance of designing massively parallel assays that can disentangle these effects. Tite-Seq provides a general solution to this problem for massively parallel studies of protein-ligand binding. Indeed, the Tite-Seq procedure described here can be readily applied to any protein binding assay that is compatible with yeast display and FACS. Many such assays have been developed (*Liu, 2015*). We expect that Tite-Seq can also be readily adapted for use with other expression platforms, such as mammalian cell display (*Forsyth et al., 2013*).

Our Tite-Seq measurements reveal interesting distinctions between the effects of mutations in the CDR1H and CDR3H regions of the anti-fluorescein scFv antibody studied here. As expected, we found that variation in and around CDR3H had a larger effect on affinity than did variation in and around CDR1H. We also found that CDR1H is more optimized for protein expression than is CDR3H, an unexpected finding that appears to be novel. Yeast display expression levels are known to correlate with thermostability (*Shusta et al., 1999*). Our data is limited in scope, and we remain cautious about generalizing our observations to arbitrary antibody-antigen interactions. Still, this finding suggests the possibility that secondary CDR regions (such as CDR1H) might be evolutionarily optimized to help ensure antibody stability, thereby freeing up CDR3H to encode antigen specificity. If this hypothesis holds, it could provide a biochemical rationale for why CDR3H is more likely than CDR1H to be mutated in functioning receptors (*Liberman et al., 2013*) and why variation in CDR3H is often sufficient to establish antigen specificity (*Xu and Davis, 2000*).

Tite-Seq can also potentially shed light on the structural basis for antibody-antigen recognition. By comparing the effects of mutations with the known antibody-fluorescein co-crystal structure (*Whitlow et al., 1995*), we identified a strong correlation between the effect that a position has on affinity and the number of molecular contacts that the residue at that position makes within the antibody. By contrast, no such correlation of expression with this number of contacts is observed. Again, we are cautious about generalizing from observations made on a single antibody. If our observation were to hold for other antibodies, however, it would suggest that the functional geometry of

paratopes might be governed by networks of residues whose positions and orientations are strongly interdependent.

## Materials and methods

Tite-Seq was performed as follows. Variant 3H and 1H regions were generated using microarray-synthesized oligos (LC Biosciences, Houston TX. USA). These were inserted into the 4-4-20 scFv of (*Boder and Wittrup, 1997*) using cassette-replacement restriction cloning as in (*Kinney et al., 2010*); see Appendix 3. Yeast display experiments were performed as previously described (*Boder et al., 2000*) with modifications; see Appendix 2. Sorted cells were regrown and bulk DNA was extracted using standard techniques, and amplicons containing the 1H and 3H variable regions were amplified using PCR and sequenced using the Illumina NextSeq platform; see Appendix 4. Three replicate experiments were performed on different days. Raw sequencing data has been posted on the Sequence Read Archive under BioProject ID PRJNA344711. Low-throughput flow cytometry measurements were performed on clones randomly picked from the Tite-Seq library. Sequence data and flow cytometry data were analyzed using custom Python scripts, as described in Appendices 5 and 6. Processed data and analysis scripts are available at github.com/jbkinney/16_titeseq.

## Acknowledgements

We would like to thank Jacklyn Jansen, Amy Keating, Lother Reich, and Bruce Stillman for helpful discussions. We would also like to thank Dane Wittrup for sharing plasmids and yeast strains. RMA, TM and AMW were supported by European Research Council Starting Grant n. 306312. JBK was supported by the Simons Center for Quantitative Biology at Cold Spring Harbor Laboratory.

## Additional information

### Funding

| Funder | Grant reference number | Author |
|---|---|---|
| European Research Council | StG n. 306312 | Rhys M Adams<br>Thierry Mora<br>Aleksandra M Walczak |
| Simons Center for Quantitative Biology | | Justin B Kinney |

The funders had no role in study design, data collection and interpretation, or the decision to submit the work for publication.

### Author contributions

RMA, Data curation, Software, Formal analysis, Validation, Investigation, Visualization, Methodology, Writing—original draft, Writing—review and editing; TM, Conceptualization, Supervision, Validation, Investigation, Methodology, Writing—original draft, Writing—review and editing; AMW, Conceptualization, Supervision, Funding acquisition, Validation, Investigation, Methodology, Writing—original draft, Writing—review and editing; JBK, Conceptualization, Supervision, Funding acquisition, Validation, Investigation, Visualization, Methodology, Writing—original draft, Writing—review and editing

### Author ORCIDs

Thierry Mora, http://orcid.org/0000-0002-5456-9361
Aleksandra M Walczak, http://orcid.org/0000-0002-2686-5702
Justin B Kinney, http://orcid.org/0000-0003-1897-3778

## Additional files

### Major datasets

The following dataset was generated:

| Author(s) | Year | Dataset title | Dataset URL | Database, license, and accessibility information |
|---|---|---|---|---|
| Adams RM, Kinney JB, Mora T, Walczak AM | 2016 | Saccharomyces cerevisiae high-throughput titration curves | https://www.ncbi.nlm.nih.gov/bioproject/344711 | Publicly available at the NCBI BioProject database (accession no: PRJNA344711) |

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

## Appendix 1

### Schematic simulations

For panels D and E of *Figure 1*, data was simulated (using *Equation 1*) for two hypothetical scFvs: one similar to WT, with $K_D = 1.2 \times 10^{-9}$ M, $A = 300$, and $B = 10$, and one similar to a typical mutant, with $K_D = 10^{-6}$ M, $A = 1000$, $B = 10$. Simulated sorts were performed at the eleven antigen concentrations used in our experiments ($c = 0$ M, $10^{-9.5}$ M, $10^{-9.0}$ M, ..., $10^{-5.0}$ M). For each clone at each antigen concentration, fluorescence signals were simulated for 1000 cells by multiplying the $f$ quantity in *Equation 1* by a factor of $\exp(\eta)$ where $\eta$ is a normally distributed random number. *Figure 1D* shows the mean values of these simulated fluorescence signals. Curves of the form in *Equation 1* were fit to these data by minimizing the square deviation between predicted $\log_{10} f$ values and the $\log_{10}$ mean of the simulated fluoresence values. The Tite-Seq measurements illustrated in *Figure 1E* were simulated by sorting 1000 cells, using fluorescence values generated in the same manner as above, into three bins defined by the following fluorescence boundaries: $(0, 30)$ for bin 0, $(30, 300)$ for bin 1, and $(300, \infty)$ for bin 2. The mean bin number for each clone at each antigen concentration was then computed. Curves having the form in *Equation 1* were then fit to these data by minimizing the square deviation of predicted $\log_{10} f$ values from these mean bin values.

## Appendix 2

### Yeast display

To help ensure consistency across samples, the yeast display cultures used in our low-throughput flow cytometry measurements and in our Tite-Seq experiments were inoculated with carefully prepared frozen liquid culture inocula. Specifically, inoculation cultures were grown at $30°C$ in SC-trp +2% glucose to an OD600 value between 0.9 and 1.1, then stored at $-80°$ in aliquots containing 10% glycerol and either 0.4 ml·OD of cells (for clones) or 1 ml·OD of cells (for libraries).

The expression of yeast-displayed scFvs was induced as follows. Liquid cultures of SC-trp +2% glucose were inoculated using single frozen inocula, yielding an approximate starting OD of 0.05. These cultures were grown at $30°C$ for 8 hr; the final OD of these cultures was approximately 0.7. Cells were then spun down at 1932 g for 8 min at $4°C$, resuspended in SC-trp +2% galactose+0.1% glucose at 0.2 OD, and incubated for 16 hr at $20°C$. We note that adding 0.1% glucose to these galactose induction cultures was essential for reliably achieving scFv expression in a large fraction of yeast cells.

Induced yeast were fluorescently labeled as follows. Galactose induction cultures were spun down and washed with ice cold TBS-BSA (0.2 mg/ml BSA, 50 mM Tris, 25 mM NaCl, pH 8). This yielded approximately 5.3 ml·OD of cells for Tite-Seq FACS. For antigen binding reactions, cells were then resuspended in a primary labeling reaction containing 40 ml TBS-BSA and biotinylated fluorescein (ThermoFisher B1370) at a concentration between 0 M and $10^{-5}$ M, then incubated with shaking for 1 hr at room temperature. Reaction volumes were large enough to ensure that $\geq$ 10 antigen molecules per scFv were present, assuming $\sim 10^5$ scFvs per cell (**_Boder and Wittrup, 1997_**). Cells were then washed twice with 40 ml ice cold TBS-BSA, suspended in a secondary labeling reaction containing 1 ml ice-cold TBS-BSA and 7 $\mu$g/ml streptavidin R-PE (ThermoFisher S866), and incubated for 30 min at $4°C$ while shaking. Cells were then spun down and resuspended in ice cold TBS-BSA and saved for FACS later that day. Expression labeling reactions proceeded in the same manner, except that the primary labeling reaction contained 1.4 $\mu$g/ml rabbit anti-c-Myc antibody (Sigma-Aldrich C3956) in place of the antigen, and the secondary labeling reaction contained 0.8 $\mu$g/ml BV421-conjugated donkey anti-rabbit antibody (BioLegend 406410) in place of streptavidin R-PE. The labeling reactions used to filter out improperly cloned scFvs (as described in Appendix 3) proceeded in the same manner as the expression labeling reaction, except that 0.8 $\mu$g/ml mouse anti-HA antibody (Roche 11583816001) was added to the primary labeling reaction, while 0.4 $\mu$g/ml APC-conjugated anti-mouse antibody (BD Biosciences 550826) was added to the secondary labeling reaction. For clonal flow cytometry measurements, we kept reagent and cell concentrations the same as described above, but reduced reaction volumes 27-fold. Secondary labeling reactions with streptavidin R-PE were done at 4 $\mu$g/ml in 112.5 $\mu$l. Secondary labeling reactions with 0.8 $\mu$g/ml BV421-conjugated donkey anti-rabbit antibody were performed in 60 $\mu$l.

## Appendix 3

### Cloning strategy

Amplicons containing variable CDR1H or CDR3H regions were generated as follows. An oligonucleotide library containing mutagenized 1H and 3H variable regions (see **Table 2**) was generated by LC Sciences using microarray-based synthesis. The specific oligos used are provided at github.com/jbkinney/16_titeseq. 1H and 3H library oligos were separately amplified via PCR using primers oRAL10 and oRAR10 (for 1H) or oRAL11 and oRAR11 (for 3H). Oligos containing the WT sequence were amplified from plasmid pCT302 (**Boder and Wittrup, 1997**) using primers 1H2F and 1H1R (for the 1H region) or 3H1F and 3H2R (for the 3H region). Overlap-extension PCR using primers oRA10 and oRA11, one oligo library (1H or 3H) and the complementary WT oligo (3H or 1H, respectively), and plasmid pCT302, were then used to create the iRA11 amplicon library (**Figure 2—figure supplement 1A**). Note that each amplicon in this library had mutations only in the 1H variable region or in the 3H variable region, but not in both of these regions.

The pRA10 cloning vector (**Figure 2—figure supplement 1B**) was assembled using Gibson cloning (**Gibson et al., 2009**) with template plasmids pCT302 (**Boder and Wittrup, 1997**) and pJK14 (**Kinney et al., 2010**). pCT302 is the yeast display expression plasmid containing the WT scFv. pJK14 contains a ccdB cloning cassette flanked by outward-facing BsmBI restriction sites. pRA10 closely resembles pCT302, except that it contains the ccdB cassette from pJK14 in place of the region of the scFv gene that we aimed to mutagenize. Multiple spurious BsmBI restriction sites present pCT302 were also removed in pRA10. pRA10 was propagated in *Escherichia coli* strain DB3.1, which is resistant to the CcdB toxin.

The pRA11 plasmid library (**Figure 2—figure supplement 1C**) was generated by digesting pRA10 with BsmBI, digesting the iRA11 amplicon library with BsaI, and subsequent ligation with T4 DNA ligase. Ligation reactions were desalted and transformed into DH10B E. coli via electroporation, yielding $\geq 10^8$ transformants. The 1H and 3H libraries were cloned separately.

The pRA11 libraries were introduced into the EBY100 strain of *Saccharomyces cerevisiae* using high-efficiency LiAc transformation (**Gietz and Schiestl, 2007**). This yielded $\geq 10^5$ transformants. To filter out yeast containing improperly cloned scFvs, we induced scFv expression, immuno-affinity labeled the HA and c-Myc epitopes on the scFv, and used FACS to recover $8 \times 10^5 - 2 \times 10^6$ cells that registered positive for both epitopes. The scFv induction and labeling procedures used to do this are described in Appendix 2. 144 yeast clones were picked at random from this library and submitted for low-throughput Sanger sequencing of the 1H and 3H variable regions of the scFv. Based on preliminary Tite-Seq experiments, 19 of these clones were then chosen for low-throughput $K_D$ measurements.

## Appendix 4

### Tite-Seq procedure

The inocula used for our Tite-Seq experiments comprised yeast harboring the 1H and 3H pRA11 plasmid libraries, mixed in equal proportions, and spiked at 0.625% with OPT-containing yeast (as a positive control) and at 0.625% with $\Delta$-containing yeast (negative control). Cells were then grown, induced, and labeled with antigen at eleven different concentrations (0 M, $10^{-9.5}$ M, $10^{-9.0}$ M, . . ., $10^{-5.0}$ M) as described in Appendix 2.

Each batch of labeled cells was then sorted, using FACS, over a period of approximately 20 min. During FACS, cells were first filtered based on forward scatter and side scatter to help ensure exactly one live cell per droplet. Cells passing these criteria were sorted into four bins based on R-PE fluorescence. The fluorescence gates used in these sorts were kept the same across all antigen concentrations (see *Figure 3*, *Figure 3—figure supplement 1*, and *Figure 3—figure supplement 2*). Cells were sorted into a rounded 5 ml polypropylene tube containing 1 ml 2X YPAD media. In our separate Sort-Seq experiments assaying scFv expression levels, cells were prepared and sorted in the same way, save for the changes to the labeling reaction described in Appendix 2 and the use of gates on BV421 fluorescence instead of R-PE fluorescence.

Each of the 48 bins of sorted cells, as well as a sample of unsorted cells, were then deposited in 5 ml of SC-trp +2% glucose and regrown overnight at $30°$C. Approximately 25 ml·OD of cells were then spun down, resuspended in a lysis reaction containing 200 $\mu$l of 0.5 mm glass beads, 200 $\mu$l of Phenol/chloroform/isoamyl alcohol and 200 $\mu$l of yeast lysis buffer (10 mM NaCl, 1 mM Tris, 0.1 mM EDTA, 0.2% Triton X-100, 0.1% SDS), and vortexed for 30 min. 200 $\mu$l of water was added, cells were spun down, and the aqueous layer was extracted. Four subsequent extractions were performed, the first two using 200 $\mu$l of Phenol/chloraform/isoamyl alcohol, the second two using 200 $\mu$l for chloroform/isoamyl alcohol. Bulk nucleic acid was then ethanol precipitated and resuspended in 100 $\mu$l of IDTE (Integrated DNA Technologies).

Two rounds of PCR were then performed on each of the 49 samples of bulk nucleic acid. In the first round of PCR, primers L1AF_XX and L2AF_XX were used to amplify the 1H-to-3H region and to add a bin-specific barcode (numbered XX = 01, 02, . . ., 64) on either end of the 1H-to-3H region; see *Figure 2—figure supplement 1*. To keep PCR crossover to a minimum, only 15 PCR cycles were used. These 49 PCR reactions were then pooled, purified using a QIAquick PCR purification kit (Qiagen), and used as template for second round of PCR with primers PE1v3ext and PE2v3. Again, to keep crossover to a minimum, only 25 PCR cycles were used. This PCR reaction was again purified, mixed with PhiX DNA (at ∼25% molarity) and submitted for sequencing using the Illumina NextSeq platform.

Analysis of the resulting sequence data across the three replicate Tite-Seq experiments revealed that some of the 147 FACS bins were highly under-sampled. This under-sampling likely resulted from the use of a non-saturating number of PCR cycles prior to pooling. The different barcodes incorporated into the PCR primers also appear to have affected amplification efficiency to different extents. To even out the distribution of reads across bins, we selected the 27 most poorly sampled bins, re-amplified the 1H-to-3H regions in these bins using primers with different barcodes than before, and submitted the resulting amplicons for a fourth round of Illumina NextSeq sequencing.

## Appendix 5

# Inference of $K_D$ from Tite-Seq data

We modeled the binding titration curve of each sequence – i.e., the curve describing how mean cellular fluorescence depends on antigen concentration – using a non-cooperative Hill function. Making the dependence on scFv sequence $s$ and antigen concentration $c$ explicit, **Equation 1** of the main text becomes

$$f_{sc} = A_s \frac{c}{c + K_{D,s}} + B, \tag{A1}$$

where $f_{sc}$ denotes the mean fluorescence of cells carrying sequence $s$ and labeled with antigen at concentration $c$, $B$ represents the autofluorescence of cells, and was set equal to the mean fluorescence of cells labeled at 0 M antigen. $A_s$ is the increase in fluorescence due to saturation of all surface-displayed scFvs, and $K_{D,s}$ is the dissociation constant for sequence $s$. We inferred $A_s$ and $K_{D,s}$ for all sequences $s$ as follows.

Tite-Seq does not provide direct measurements of the fluorescence $f_{sc}$. Instead, we approximated this quantity using a weighted averaged over sorting bins. Specifically, we assumed that

$$\ln f_{sc} \approx \sum_b p_{b|sc} F_{bc}. \tag{A2}$$

Here, $F_{bc}$ is the mean log fluorescence of the cells that were sorted at concentration $c$ into bin $b$, and $p_{b|sc}$ is the probability that a cell having sequence $s$ and labeled at concentration $c$, if sorted, would be found in bin $b$. Values for $F_{bc}$ were computed directly from the FACS data log. The probabilities $p_{b|sc}$, by contrast, were inferred from Tite-Seq read counts. These probabilities are closely related to $R_{bsc}$, the number of sequence reads for each sequence $s$ from bin $b$ at antigen concentration $c$. This relationship is complicated by additional factors that arise from variability in sequencing depth from bin to bin. Moreover, because there were often a small number of reads for any particular sequence in a given bin, it was necessary in our inference procedure to treat the relationship between $p_{b|sc}$ and $R_{bsc}$ probabilistically.

We therefore inferred values for the probabilities $p_{b|sc}$ through the following maximum likelihood procedure. First, we assumed that the number of reads $R_{bsc}$ is related to an 'expected' number of reads $r_{bsc}$ via a Poisson distribution. The log likelihood of observing a specific set of read counts $R_{bsc}$ over all bins $b$ and concentrations $c$ for a given sequence $s$ is therefore given by

$$L_s = \ln \left[ \prod_{b,c} \frac{1}{R_{bsc}!} r_{bsc}^{R_{bsc}} e^{-r_{bsc}} \right]. \tag{A3}$$

The expected number of reads $r_{bsc}$ is, in turn, related to the probability $p_{b|sc}$ via

$$r_{bsc} = \frac{R_{bc}}{C_{bc}} C_c P_s p_{b|sc}. \tag{A4}$$

Here, $R_{bc} = \sum_s R_{bsc}$ is the total number of reads from bin $b$ at antigen concentration $c$, $C_{bc}$ is the number of cells sorted into bin $b$ at antigen concentration $c$ (obtained from the FACS data log), $C_c = \sum_b C_{bc}$ is the total number of cells sorted at concentration $c$, and $P_s$ is the fraction of cells in the library with sequence $s$. The factor $R_{bc} C_c / C_{bc}$ in **Equation A4** accounts

for differences in the depth with which each bin was sequenced. Note: **Equation A3** assumes that each final read arose from a different sorted cell. This assumption is clearly violated if $R_{bc} > C_{bc}$. In cases where this inequality was found to hold, we rescaled all $R_{bsc} \rightarrow hR_{bsc}$ where $h = C_{bc}/R_{bc}$ before undertaking further analysis.

For each sequence $s$, we inferred $K_{D,s}$, $A_s$, $P_s$, and all probabilities $p_{b|sc}$ by maximizing the likelihood $L_s$ subject to the constraint that

$$\sum_b p_{b|sc} F_{bc} = \ln\left(A_s \frac{c}{c + K_{D,s}} + B\right) \tag{A5}$$

at every concentration $c$. Note that, in this procedure, the sorting probabilities $p_{b|sc}$ are not modeled explicitly as a function of the putative mean fluorescence. Instead, they are inferred from the data along with the parameters of the non-cooperative Hill function. Doing this dispenses with the need for a detailed characterization of the noise in the Tite-Seq sorting procedure. The validity of this procedure is evinced by the analysis of simulated data, shown in **Figure 4—figure supplement 8**.

The maximum likelihood optimization problem described above was solved as follows. For each concentration $c$, we created a grid of 100 equally spaced points for $\ln f_{sc} \in [F_{0a}, F_{3a}]$. For each possible value of $f_{sc}$, we then used Nelder-Mead optimization of $p_{b|sc}$ to minimize $L_s$ under the constraint in **Equation A5**. Akima interpolation was then used to create a function of the optimized probability $\hat{p}_{b|sc}$ as a function of $f_{sc}$. We then scanned a $321 \times 201$ grid of values for the pair $(K_{D,s}, A_s)$ and selected the pair that minimized $L_s(K_{D,s}, A_s, \{\hat{p}_{b|sc}(K_{D,s}, A_s), P_s\}_{b,c})$. We repeated this scan by varying $P_s$ over 95 different values. The final inferred values for $K_{D,s}$, $A_s$, and $P_s$ were those so found to maximize $L_s$. Python code for this inference procedure is provided at github.com/jbkinney/16_titeseq.

## Appendix 6

### Inference of $K_D$ from flow cytometry experiments on individual clones

Low-throughput flow cytometry measurements performed on clonal cell populations were used to measure $f_{sc,\text{flow}}$, the mean fluoresence of cells carrying sequence $s$ and labeled at antigen concentration $c$. As for the Tite-Seq inference procedure described in Appendix 5, it was assumed that $f_{sc,\text{flow}}$ could be modeled using the non-cooperative Hill function $A_s c/(c + K_{D,s}) + B$, where $A_s$ is the increase in mean fluorescence due to fully labeled scFvs of sequence $s$, $K_{D,s}$ is the corresponding dissociation constant, and $B$ is background fluorescence. $B$ was computed from the average fluorescence of clones at 0 M fluorescein. $A_s$ and $K_{D,s}$ were then inferred by minimizing the square deviation between measured $\ln f_{sc,\text{flow}}$ values and log Hill function predictions, i.e.,

$$\sum_c \left[ \ln f_{sc,\text{flow}} - \ln \left( A_s \frac{c}{c + K_{D,s}} + B \right) \right]^2. \tag{A6}$$

This optimization procedure was performed using a grid search algorithm in which $K_{D,s}$ was restricted to the interval $[10^{-10} M, 10^{-3} M]$ and $A_s$ was restricted to the interval $[\hat{A}, 100\hat{A}]$ where $\hat{A}$ denotes the average range of fluorescence values over the 4-to-8 clones assayed per flow cytometry session.

## Appendix 7

### Realistic Tite-Seq simulations

In order to test our analysis pipeline, we simulated realistic Tite-Seq data and analyzed it with the same scripts that we used on real data. For each sequence $s$, a $K_{D,s}$ value was randomly drawn from the interval $[10^{-10}$ M, $10^{-4}$ M$]$ using a uniform distribution in log space, and an $A_s$ value was drawn uniformly from a uniform distribution spanning the bulk of experimentally observed $A$ values. At each of the eleven antigen concentrations $c$, we then modeled the distribution of cellular fluorescence using a Gaussian Mixture Model (GMM) in log space. Specifically, letting $x$ denote $\log_{10}$ cellular fluorescence values, we assumed that the probability density describing $x$ to be

$$P_{sc}(x) = \frac{\alpha}{\sigma_0\sqrt{2\pi}}e^{-\frac{(x-\mu_0)^2}{2\sigma_0^2}} + \frac{1-\alpha}{\sigma_1\sqrt{2\pi}}e^{-\frac{(x-\mu_{sc})^2}{2\sigma_1^2}}, \tag{A7}$$

where $\alpha = 0.2$ is the fraction of non-expressing cells, $\mu_0 = 4.77$ and $\sigma_0 = 1$ are the mean and standard deviation of $x$ values for dark cells, and $\sigma_1 = 0.5$ is the standard deviation of $x$ values for scFv-expressing cells. The mean $x$ value of scFv-expressing cells, here denoted $\mu_{cs}$, was chosen so that the population average of $x$ is given by the Hill function in **Equation A1**, i.e., so that

$$\langle x \rangle_{P_{sc}} = A_s\frac{c}{c+K_{D,s}} + B. \tag{A8}$$

The left hand side of **Equation A8** can be computed analytically using **Equation A7**. Doing this and solving for $\mu_{sc}$ gives

$$\mu_{sc} = \ln\left(\frac{A_s c}{c+K_{D,s}} + B - \alpha e^{\mu_0+\sigma_0^2/2}\right) - \ln(1-\alpha) - \frac{\sigma_1^2}{2}; \tag{A9}$$

this is the specific formula we used to compute $\mu_{sc}$ as a function of $c$, $A_s$, and $K_{D,s}$. Next we computed exact $p_{b|sc}$ values using

$$p_{b|sc} = \int_{b^-}^{b^+} dx\, P_{sc}(x), \tag{A10}$$

where $b^+$ and $b^-$ are the upper and lower fluorescence bounds used for bin $b$ in our Tite-Seq experiment (replicate number 1). These values were then used to draw read counts $R_{bsc}$ for each sequence $s$ values via

$$R_{bsc} \sim \text{Binomial}(n = k_s R_{bc}, p = p_{b|sc}), \tag{A11}$$

where $k_s$ is a random variable, uniformly distributed on a log scale between 0.01 and 100, used to represent noise due to PCR jackpotting. Data thus simulated for WT values of $K_D$ and $A$ are shown in **Figure 4—figure supplement 7**.

