## [Decision Letter]

[Editors’ note: a previous version of this study was rejected after peer review, but the authors submitted for reconsideration. The first decision letter after peer review is shown below.]

Thank you for submitting your work entitled "Measuring the sequence-affinity landscape of antibodies with massively parallel titration curves" for consideration by *eLife*. Your article has been reviewed by three peer reviewers, and the evaluation has been overseen by a Reviewing Editor and Aviv Regev as the Senior Editor. The following individuals involved in the review of your submission have agreed to reveal their identity: Claudia Bank and Dmitriy Chudakov (peer reviewers).

I regret that after careful discussion with the reviewers, we have come to the decision to reject the manuscript in its current form.

All of us believe that your new approach is extremely exciting and potentially powerful. However, the reviewers identified a number of shortcomings that led us to decide against recommending acceptance of the manuscript. In particular, the manuscript as written clearly is fundamentally a methodology paper rather than a paper reporting a novel new biological result. However, many of the standards necessary to establish the robustness and reproducibility of a new method are not adequately met.

The policy of *eLife* is to not require revisions that would involve large amounts of new work, and as you will see below it would require substantial new work to address the concerns raised by the reviewers. We recognize that you may therefore either prefer to submit the paper elsewhere, or make extensive changes and re-submit to *eLife*. Below I outline what I consider to be the main points so that you are aware of them if you choose the latter route. In addition, the full reviewer comments are pasted below.

An experimental replicate is needed to assess reproducibility of the method. While we appreciate the effort to develop an error model using synonymous counts, for a new high-throughput methodology, the only robust ways to assess errors is to perform an independent replicate of the experiment and assess the variation between replicates. Replicates are a standard in recent deep mutational scanning studies (e.g. PMID 25723163, 25006036), and we would require one before accepting the paper. It is not essential that the replicates have perfect correlations (we don't expect that they will), but it is necessary to have some independent means of assessing the noise and validating which findings are robust to this noise.

Reviewer 2's point about what "experimental details" constrained the use of ligand values outside the *K_D_* range must be addressed – is the method inapplicable to antibodies with higher affinities? If so, that would be a very important limitation. This same point applies to the low-throughput assays. In the original Wittrup paper (cited in this manuscript), low-throughput replicates yielded *K_D_* estimates that varied by no more than 2-fold. Here, the *K_D_* estimates vary by two orders of magnitude. The reason appears to be that the range of ligand concentrations does not contain the *K_D_*, which is a basic requirement of a proper titration curve.

*eLife*'s policy is to encourage making data and computer code available. We encourage you to do this regardless of whether you re-submit to *eLife* or elsewhere.

The reviews list a variety of other easy-to-fix points, such as better citations of the literature and clarification of unclear points.

*Reviewer #1:*

In this manuscript the authors present a new method called Tite-Seq to assess the effect of mutations in an antibody on antigen-binding affinity in a high-throughput fashion. They use their method on two regions of the scFv antibody and report a correlation between the number of contacts of a wt residue inside the antibody and its sensitivity to mutations.

Although my expertise in this area is limited, I am generally fascinated by novel high-throughput approaches, and I believe that the presented approach may prove useful to study antigen binding affinities on a large scale. However, I have several concerns regarding the validation of the approach and the appeal of the biological results.

Most importantly, I would like to see a true replicate experiment in order to get an idea of the correlation between measured values on a larger scale than just a couple of low-throughput comparisons. As far as I know, this is common practice when presenting a new approach like this, and it would (1) allow for a better idea of the error (which, in my opinion, is calculated in a highly optimistic way), and (2) allow for a much better quantification of the results. E.g., is the difference of 56% vs. 41% of mutations above the detection limit (in subsection “E. Differing effects of mutations in CDR1H and CDR3H”, first paragraph) a large one, or maybe not even distinguishable given the accuracy of the experiment? Even if the same library was used, a replicate of the subsequent steps would be highly informative, and, in my opinion, necessary for validation of the approach.

I do not see a striking biological result from the analysis. What occurs to me as the main result of the paper (the correlation between contacts of wt residue with sensitivity to mutations at that position) is not too surprising to me, given many studies that have shown protein stability to be an important determinant of its function (and, as I understand, the sensitivity is measured on an absolute scale). Other reported findings seem suspicious and vague, especially considering my main concern expressed above.

As I understand, 1850 mutations per region were surveyed. Such a high number introduces a lot of noise due to sampling by FACS and sequencing, even if the initial library is large and evenly distributed (and especially if the number of sequences is not large for some data points, cf. Figure 2—figure supplement 2B). I may have missed this, but is the distribution of the mutations in the initial library known, and what was the distribution of absolute reads per mutant at each data point?

*Reviewer #2:*

The authors present Tite-Seq, a method that uses high-throughput DNA sequencing of yeast-displayed antibody libraries to assess mutant *K_D_* and surface expression at a large scale. In general, the experiments and analysis are well executed, and the paper is mostly well written. My enthusiasm is restrained for two main reasons. First, the authors fail to cite and discuss substantially similar work. Their elucidation of *K_D_* values for a large library of variants does represent an advance, but it's incremental. The work must be discussed in the context of what has come before. Second, several of the analyses are poorly described and, if I understood them correctly, will inflate the reader's confidence in the method. In particular, the method for estimating error for each *K_D_* value is insufficient. Both of these points are particularly important because the manuscript is focused on the method itself rather than an important/exciting application. I therefore do not recommend publication of the manuscript in its current form.

Specific comments:

Introduction – The authors should be more thorough in discussing the strengths and weaknesses of prior deep mutational scanning work. In particular, the fact that yeast display has been coupled to deep mutational scanning and that affinity ranking of variants has been achieved is something readers should know before they get to the end of the Discussion (PMID 25311858). Mammalian antibody display has also been coupled to deep mutationals canning (PMID 23765106). Ribosome antibody display and deep mutational scanning using varying ligand concentrations has also been done (PMID 23103372). Of course, the existence of a substantially similar paper (PMID 26296891) should also be acknowledged and the work discussed in that context.

Subsection “C. Low-throughput validation experiments” – The authors show low-throughput validation data for three clones and compare this data to Tite-Seq results. In the subsection “A. Overview of Tite-Seq”, the authors make the point that, if curve fitting is to be successful, the *K_D_* value must fall within the range of ligand values used. However, for the validation data shown, the fit *K_D_* values are generally at the very low end of the ligand concentration range. In Figure 3, most of the titration curves more closely resemble flat lines. This can be explained by the fact that the *K_D_* of the WT is 0.7e-9 M whereas the lowest concentration of ligand used was 1e-8.5 M. I am puzzled as to why the authors chose to employ a ligand concentration range that did not include the known *K_D_* of the WT sequence. In the Discussion they mention "experimental details," that constrained them, but I'd suggest a fuller and earlier disclosure of these limitations. Given that most antibody-antigen interactions have *K_D_s* at least as low as the one studied, I wonder whether the method could really be generally applied.

Subsection “C. Low-throughput validation experiments” – Pursuant to the previous comment, the authors' simulations (Figure 1) are done using two hypothetical interactions whose *K_D_s* are substantially higher than the experimental case considered. They should do simulations using the actual *K_D_* and ligand concentrations employed.

Subsection “C. Low-throughput validation experiments” – The low-throughput validation experiments do not themselves look particularly robust. The *K_D_* values inferred from these experiments range over two orders of magnitude. The authors state "Although individual data points can be noisy, fitting curves to multiple data points nevertheless provide reasonably accurate measurements of affinity. The accuracy of these measurements is increase by averaging over replicates." This is a vague statement. How noisy? How much does replication help? A key problem, I think, is that the low throughput flow validation data looks as noisy as the Tite-Seq data. These experiments should be improved and repeated. Even better, an orthogonal method of validation should be employed.

Figure 2 – This panel and accompanying figure legend is vague. The text makes it clear that the numbers show the diversity of each library, but it would be helpful to clarify the legend.

Figure 4/subsection “D. Tite-Seq can measure dissociation constants” – In the figure legend the authors state "Error bars on flow *K_D_* values are the same for all data points; they show the average mean squared error computed computed using three replicate measurements for each clone." In the text, they state "Error bars on flow cytometry *K_D_* values were computed using the average variance observed in replicate measurements." I have a few problems here. First, neither statement makes totally clear what the authors did/what is shown in Figure 4. Furthermore, MSE usually used when the parametric value of an estimator is known. For the flow data, we don't have parametric values, just three replicates. So, a confidence interval would be more appropriate. Finally, and most troublingly, the authors have elected to present the mean variance/MSE of all three clones rather than the variance (or CI, or whatever) of each clone independently. I can't think of a good reason to do this. Because the WT replicates happened to look great while the two clones didn't, using the mean of all three is misleading. The authors should improve this analysis and then better explain it. Also, note the repeated "computed."

Figure 4/Subsection “D. Tite-Seq can measure dissociation constants” – The error estimation procedure for Tite-Seq, based on a single experiment, uses synonymous variants to estimate the standard deviation of each measurement. Variants are binned based on read depth and then a regression is performed to determine the error in each bin. If I understood the procedure, the error is assumed to be wholly dependent on read depth. The read depth vs. signal to noise/error plot (Figure 4) is really noisy, which suggests that the read depth alone does not capture all of the error. Many factors are known to impact experiment-to-experiment variability (e.g. PCR bias, experimenter error, etc.) in these types of experiments. Replication would enable much more robust error estimation, and would considerably strengthen the work.

Subsection “D. Tite-Seq can measure dissociation constants” – The authors state: "We note that…measurements for the WT scFV are about a factor of 10 larger than the previously measured value of *K_D_*". I wonder if this is due to the lack of data points at lower concentrations rather than any differences in buffer/etc. Again, a gold-standard, non flow-based assay would help here.

Figure 6 – Because of the two-dimensional heatmap it is somewhat difficult to appreciate the relationship between expression and binding. Two separate panels with individual color schemes might be easier to comprehend.

Appendix C – The authors say they used custom microarray nucleotides to generate the library, but the nature of the sequences on the array is unclear. A supplementary data file containing the sequences ordered should be included.

Appendix F – The authors used the 10% lowest fluorescence values to estimate autofluorescence/background. This is a curious choice, if the library contained mutations to stop codons (see comment above). Stop codons, particularly early ones, virtually guarantee loss of function.

*Reviewer #3:*

I could hardly evaluate the mathematical aspects of the work. However, it looks like the previous expertise of this team ensures the high mathematical quality.

Concerning the whole idea of the work – I believe it is brilliant.

The approach allows studying the aspects of dependence of antibody affinity and potentially cross-reactivity (with several antigens used) on the sequence landscape, which is beautiful.

[Editors’ note: what now follows is the decision letter after the authors submitted for further consideration.]

Thank you for submitting your heavily revised manuscript "Measuring the sequence-affinity landscape of antibodies with massively parallel titration curves" for consideration by *eLife*. The manuscript was evaluated by three reviewers who are experts in the field: two were also reviewers for the original submission, and one is a new reviewer. The evaluation has been overseen by Jesse Bloom as the Reviewing Editor and Aviv Regev as the Senior Editor. The following individuals involved in review of your submission have agreed to reveal their identity: Timothy A Whitehead (Reviewer #1) and Claudia Bank (Reviewer #2).

We anticipate that we can accept your manuscript for publication provided that you make revisions that address the comments below.

First, we appreciate your careful attention to addressing the issues raised in the in the initial review, particularly adding replicates and extending the antigen range. We recognize that these were time-consuming changes, and the seriousness with which you took those critiques has greatly improved this paper. Your study now represents an impressive use of deep sequencing to obtain fairly rigorous *K_D_* values.

Specific points:

1) Please make sure that your deep sequencing data is available on the SRA and relevant computer code is available as supporting file or on a publicly accessible repository. If this is already the case, please clearly indicate in the manuscript where these can be found.

2) There are a few additional papers that you should consider discussing in the context of prior work:

Doolan KM, Colby DW: Conformation-dependent epitopes recognized by prion protein antibodies probed using mutational scanning and deep sequencing. J. Mol. Biol. 2015, 427:328-340.

Van Blarcom T, Rossi A, Foletti D, Sundar P, Pitts S, Bee C, Melton Witt J, Melton Z, Hasa-Moreno A, Shaughnessy L, et al.: Precise and efficient antibody epitope determination through library design, yeast display and next-generation sequencing. J. Mol. Biol. 2015, 427:1513-1534.

3) Your approach estimates fairly accurate *K_D_* at the cost of more experiments (multiple binds, multiple expression levels). For certain applications where precise *K_D_* values are overkill, it may be more efficacious to use the simpler designs like those used in some of the references that you cite – perhaps worth mentioning. Also, do you have any comments on the trade-off between sequencing depth and the accuracy of the inferred *K_D_*? Similarly for the number of sorting bins? Right now it isn't clear how these were chosen. Clearly your choices worked fine, but it would be nice to explain if there was rigorous rationale for choosing these (alternatively, you could just say that exploring the effects of these parameters is interesting for future work).

4) The relatively large unsigned error on two variants in Figure 4 (approximately 2 orders of magnitude in *K_D_*) should be commented on. Why is there such a discrepancy (Subsection “D. Tite-Seq can measure dissociation constants”; Figure 4)?

5) Are the mutations outside of the dynamic range (*K_D_* > 10 μM) being used in determining correlation coefficients (Figure 4; Figure 4—figure supplement 1; Figure 4—figure supplement 2)?

6) Please quantify the error in precision and accuracy of the method. In particular, the poorer correspondence between replicates in the binding range between 10-1000 nM *K_D_* should be mentioned.

7) Can you quantify error within replicates by inferring *K_D_* for synonymous mutations?

8) How big is the benefit of controlling for active surface-displayed protein? While it is clear that yeast surface expression and folded surface displayed protein varies between variants (Burns et al., 2014), it has been shown (yet still surprising!) that surface expression (and proper folding) effects are modest for yeast surface display experiments, particularly for residues that are reasonably surface-exposed (Kowalsky and Whitehead, 2016).

Burns, Michael L., et al. Directed evolution of brain-derived neurotrophic factor for improved folding and expression in *Saccharomyces cerevisiae*. Applied and environmental microbiology 2014, 80:5732-5742.

Kowalsky CA, Whitehead TA Determination of binding affinity upon mutation for type I dockerin-cohesin complexes from Clostridium thermocellum and Clostridium cellulolyticum using deep sequencing. PROTEINS 2016 84: 1914-1928

---

## [Author Response]

[Editors’ note: the author responses to the first round of peer review follow.]

*All of us believe that your new approach is extremely exciting and potentially powerful. However, the reviewers identified a number of shortcomings that led us to decide against recommending acceptance of the manuscript. In particular, the manuscript as written clearly is fundamentally a methodology paper rather than a paper reporting a novel new biological result. However, many of the standards necessary to establish the robustness and reproducibility of a new method are not adequately met. The policy of eLife is to not require revisions that would involve large amounts of new work, and as you will see below it would require substantial new work to address the concerns raised by the reviewers. We recognize that you may therefore either prefer to submit the paper elsewhere, or make extensive changes and re-submit to eLife. Below I outline what I consider to be the main points so that you are aware of them if you choose the latter route. In addition, the full reviewer comments are pasted below. An experimental replicate is needed to assess reproducibility of the method. While we appreciate the effort to develop an error model using synonymous counts, for a new high-throughput methodology, the only robust ways to assess errors is to perform an independent replicate of the experiment and assess the variation between replicates. Replicates are a standard in recent deep mutational scanning studies (e.g. PMID 25723163, 25006036), and we would require one before accepting the paper. It is not essential that the replicates have perfect correlations (we don't expect that they will), but it is necessary to have some independent means of assessing the noise and validating which findings are robust to this noise. Reviewer 2's point about what "experimental details" constrained the use of ligand values outside the KD range must be addressed – is the method inapplicable to antibodies with higher affinities? If so, that would be a very important limitation. This same point applies to the low-throughput assays. In the original Wittrup paper (cited in this manuscript), low-throughput replicates yielded KD estimates that varied by no more than 2-fold. Here, the KD estimates vary by two orders of magnitude. The reason appears to be that the range of ligand concentrations does not contain the KD, which is a basic requirement of a proper titration curve. eLife's policy is to encourage making data and computer code available. We encourage you to do this regardless of whether you re-submit to eLife or elsewhere. The reviews list a variety of other easy-to-fix points, such as better citations of the literature and clarification of unclear points.*

We thank the editors for considering our work and for providing this encouraging assessment. We also thank the referees for their careful reading of our work and for providing thoughtful criticism. This excellent feedback has spurred us to perform additional experiments and to modify our analysis methods. We believe that these changes have greatly improved our paper. In particular,

1) The revised manuscript now describes three replicate Tite-Seq experiments. These replicates are used to quantify the uncertainty in Tite-Seq measurements.

2) We have shifted the range of antigen concentrations lower by one decade, thereby bringing the *K_D_* of the wild type antibody into experimental range. In doing so, we obtained a *K_D_* for the wild type antibody that is consistent with previous studies.

3) We have increased the number of low-throughput *K_D_* measurements used to test the validity of the affinity values found by Tite-Seq.

4) We have modified the yeast display protocol to substantially improve the fraction of antibody-displaying cells and thereby increase the precision of both our Tite-Seq and our low-throughput measurements.

5) We have implemented an improved computational method for extracting *K_D_* values from raw Tite-Seq data. The code for implementing this new method is freely available at https://github.com/jbkinney/16_titeseq

6) We have validated our new analysis method using realistic simulations, which are described in the revised text.

We believe that these changes will address the concerns voiced by the editors and referees. Below we respond to specific referee comments in more detail.

*Reviewer #1:*

*In this manuscript the authors present a new method called Tite-Seq to assess the effect of mutations in an antibody on antigen-binding affinity in a high-throughput fashion. They use their method on two regions of the scFv antibody and report a correlation between the number of contacts of a wt residue inside the antibody and its sensitivity to mutations.*

*Although my expertise in this area is limited, I am generally fascinated by novel high-throughput approaches, and I believe that the presented approach may prove useful to study antigen binding affinities on a large scale. However, I have several concerns regarding the validation of the approach and the appeal of the biological results.*

*Most importantly, I would like to see a true replicate experiment in order to get an idea of the correlation between measured values on a larger scale than just a couple of low-throughput comparisons. As far as I know, this is common practice when presenting a new approach like this, and it would (1) allow for a better idea of the error (which, in my opinion, is calculated in a highly optimistic way), and (2) allow for a much better quantification of the results. E.g., is the difference of 56% vs. 41% of mutations above the detection limit (in subsection “E. Differing effects of mutations in CDR1H and CDR3H”, first paragraph) a large one, or maybe not even distinguishable given the accuracy of the experiment? Even if the same library was used, a replicate of the subsequent steps would be highly informative, and, in my opinion, necessary for validation of the approach.*

We thank the referee for this thoughtful suggestion. In retrospect, we completely agree that replicate experiments are needed to provide appropriate validation of assays such as this. Our revised manuscript describes the results of three independent replicate Tite-Seq experiments, which are used to estimate errors, and whose results are given in Figure 4—figure supplement 2. We expect that these replicate experiments, and the analysis thereof, will address the referee’s concern.

*I do not see a striking biological result from the analysis. What occurs to me as the main result of the paper (the correlation between contacts of wt residue with sensitivity to mutations at that position) is not too surprising to me, given many studies that have shown protein stability to be an important determinant of its function (and, as I understand, the sensitivity is measured on an absolute scale). Other reported findings seem suspicious and vague, especially considering my main concern expressed above.*

Our paper is primarily a methods paper directed at addressing a fundamental problem in deep mutational scanning (DMS) assays: how to separate out the sequence-dependence of ligand binding affinity from the other sequence-dependent effects, e.g. on expression level or on the fraction of expressed proteins that are properly folded. Our work is the first to solve this fundamental problem inherent to DMS assays. These results are important because DMS assays are being rapidly adopted in a wide variety of fields including protein science, immunology, virology, and evolution.

We have also clarified the biological importance of our specific findings. Namely, our results suggest (a) that secondary CDRs many serve to stabilize antibodies against destabilizing variation in CDR3H, and (b) that binding affinity and specificity might be controlled by “sectors” within the antibody structure. Because our paper is primarily about a method, we do not follow up on these hypotheses, and so they remain preliminary. Still, this illustrates the kinds of questions that Tite-Seq can help address.

*As I understand, 1850 mutations per region were surveyed. Such a high number introduces a lot of noise due to sampling by FACS and sequencing, even if the initial library is large and evenly distributed (and especially if the number of sequences is not large for some data points, cf. Figure 2—figure supplement 2B). I may have missed this, but is the distribution of the mutations in the initial library known, and what was the distribution of absolute reads per mutant at each data point?*

We apologize for the lack of clarity on this issue. We did indeed sequence the unsorted library. Zipf plots illustrating the prevalence of each sequence in the library used for each replicate are now shown in Figure 4—figure supplement 3. Moreover, Figure 3 shows the number of cells sorted into each bin, as well as the number of reads obtained from each bin.

There is indeed a large variation in the number for reads from bin to bin, but our improved analysis method explicitly accounts for this variability. The variation in *K_D_* values measured by Tite-Seq that we expect to result from the finite sampling of each sequence was estimated using simulations. The results of these simulation tests are shown in Figure 4—figure supplement 7.

*Reviewer #2: The authors present Tite-Seq, a method that uses high-throughput DNA sequencing of yeast-displayed antibody libraries to assess mutant KD and surface expression at a large scale. In general, the experiments and analysis are well executed, and the paper is mostly well written. My enthusiasm is restrained for two main reasons. First, the authors fail to cite and discuss substantially similar work. Their elucidation of KD values for a large library of variants does represent an advance, but it's incremental. The work must be discussed in the context of what has come before. Second, several of the analyses are poorly described and, if I understood them correctly, will inflate the reader's confidence in the method. In particular, the method for estimating error for each KD value is insufficient. Both of these points are particularly important because the manuscript is focused on the method itself rather than an important/exciting application. I therefore do not recommend publication of the manuscript in its current form.*

We thank the reviewer for this critique. We believe the revised manuscript addresses all of these concerns.

Our revised manuscript provides an expanded discussion of the importance of our work in the context of the prior literature, including the works cited by the referee. We argue that our work represents an important conceptual and technological advance over previously described DMS experiments.

Specifically, all DMS experiments face an important challenge: how to separate the effect that protein sequence has on one specific biochemical property of interest from the effect that it has on other biochemical properties of the protein that can affect measurements in a DMS assay. In our case, the challenge is to separate the effect that protein sequence has on ligand binding energy from the effect that protein sequence has on protein expression and on the fraction of this expressed protein that is properly folded. Our paper solves this problem in the context of antibody-antigen interactions, and appears to be the first in the literature to do so for any protein-ligand binding energy. More generally, we provide a template for how such distinctions between sequence-function relationships can be made. The Discussion has been revised to emphasize this important point.

We agree with the referee’s criticism regarding the error bars on the Tite-Seq *K_D_* measurements. In response, we have performed the Tite-Seq experiment in triplicate, and now use these triplicate measurements to assess error bars. Our Tite- Seq measurements are also validated by an increased number of *K_D_* measurements on individual clones sampled from the library.

*Specific comments:*

*Introduction – The authors should be more thorough in discussing the strengths and weaknesses of prior deep mutational scanning work. In particular, the fact that yeast display has been coupled to deep mutational scanning and that affinity ranking of variants has been achieved is something readers should know before they get to the end of the Discussion (PMID 25311858). Mammalian antibody display has also been coupled to deep mutationals canning (PMID 23765106). Ribosome antibody display and deep mutational scanning using varying ligand concentrations has also been done (PMID 23103372). Of course, the existence of a substantially similar paper (PMID 26296891) should also be acknowledged and the work discussed in that context.*

We have expanded the Introduction to provide a more detailed discussion of the prior literature, particularly regarding prior DMS experiments using yeast display, prior work on antibody sequence-affinity landscapes, and prior work at varying ligand concentrations. The revised text now explicitly explains how our work differs substantially from these previous papers, including the papers cited by the referee above. We believe this more detailed discussion greatly clarifies why our work provides a major advance over the prior literature.

We have moved our discussion of Reich et al., 2015 (PMID 25311858), which describes the yeast-display DMS method SORTCERY, to the Introduction. We have also revised the Introduction to emphasize that both yeast display (Reich et al., 2015, PMID 25311858; Kowalsky et al., 2015, PMID 26296891) and the conceptually similar method of mammalian cell display (Forsyth et al., 2013, PMID 23765106) have indeed already been used to perform DMS experiments. We emphasize that our key advance in this regard is not the use of a cellular display system to do DMS experiments, but rather the ability to measure binding titration curves. None of these works measures binding titration curves. Therefore, all of the DMS-measured affinities reported in these three works are vulnerable to being convolved with sequence-dependent effects on expression or protein stability. Our paper shows how to overcome this common problem.

The referee is correct that Fujino et al., 2012, PMID 23103372 performed Ribosome-display-based DMS experiments at multiple concentrations. The revised text points this out. The multiple concentration measurements used by Fujino et al., however, were not used to infer binding titration curves nor to make estimates of *K_D_*; they were used only to identify codons to randomize in a combinatorial library. The only *K_D_* values reported in this paper were measured in a low-throughput manner, either by SPR or KinExA.

*Subsection “C. Low-throughput validation experiments” – The authors show low-throughput validation data for three clones and compare this data to Tite-Seq results. In the subsection “A. Overview of Tite-Seq”, the authors make the point that, if curve fitting is to be successful, the K_D_ value must fall within the range of ligand values used. However, for the validation data shown, the fit K_D_ values are generally at the very low end of the ligand concentration range. In Figure 3, most of the titration curves more closely resemble flat lines. This can be explained by the fact that the K_D_ of the WT is 0.7e-9 M whereas the lowest concentration of ligand used was 1e-8.5 M. I am puzzled as to why the authors chose to employ a ligand concentration range that did not include the known K_D_ of the WT sequence. In the Discussion they mention "experimental details," that constrained them, but I'd suggest a fuller and earlier disclosure of these limitations. Given that most antibody-antigen interactions have K_D_s at least as low as the one studied, I wonder whether the method could really be generally applied.*

We chose our initial range of concentrations to target the *K_D_s* of the bulk of sequences in our library. It does make sense, however, to include the WT *K_D_*. The triplicate experiments reported in our revised manuscript have therefore been performed with a range of antigen concentrations (10^-9.5^ M to 10^-5^ M) that is one decade lower than the range used for the previous manuscript (10^-8.5^ M to 10^-4^ M). We also modified the yeast display protocol to improve the fraction of yeast that express antibody on their surface, and this has further improved the precision of both Tite-Seq and our low-throughput flow cytometry measurements.

From our new experiments we find a WT *K_D_*of 1.9 nM using Tite-Seq and 2.4 nM using low-throughput flow cytometry. This is now consistent with the measurements from Dane Wittrup’s laboratory. We note that, although Boder et al. (2000, PMID 10984501) report a WT *K_D_*of 0.4-0.7 nM, later work (Gai and Wittrup, 2007, PMID 17870469) revised this *K_D_*upwards to 1.2 nM, which is within a factor of 2 of both our low-throughput and high-throughput measurements. The difference in salt concentration used in our experiments may account for some of the remaining discrepancy.

*Subsection “C. Low-throughput validation experiments” – Pursuant to the previous comment, the authors' simulations (Figure 1) are done using two hypothetical interactions whose K_D_s are substantially higher than the experimental case considered. They should do simulations using the actual K_D_ and ligand concentrations employed.*

The revised version of Figure 1 now shows results simulated using *K_D_*values that were measured. We emphasize in the revised caption that this figure is only meant to provide a schematic illustration of the inference method, and that the method for inferring *K_D_*values from real read counts is substantially more involved. Figure 4—figure supplement 6 and 7 illustrate more realistic simulations as well as the results that our real analysis pipeline produces from these data.

*Subsection “C. Low-throughput validation experiments” – The low-throughput validation experiments do not themselves look particularly robust. The K_D_ values inferred from these experiments range over two orders of magnitude. The authors state "Although individual data points can be noisy, fitting curves to multiple data points nevertheless provide reasonably accurate measurements of affinity. The accuracy of these measurements is increase by averaging over replicates." This is a vague statement. How noisy? How much does replication help? A key problem, I think, is that the low throughput flow validation data looks as noisy as the Tite-Seq data. These experiments should be improved and repeated. Even better, an orthogonal method of validation should be employed.*

We have modified the yeast display protocol, substantially increasing the fraction of yeast cells that display antibody. This, in turn, has substantially increased the precision of our low-throughput measurements. The low-throughput titration curves measured using this modified protocol are described in the revised manuscript.

We have also quantified the uncertainty in all high-throughput and low-throughput *K_D_*measurements. High-throughput measurements, in triplicate, with estimated error bars, are provided as the supplemental data files. A new Supplementary file I shows both high-throughput and low-throughput *K_D_*measurements for each of the clones plotted in Figure 4.

*Figure 2 – This panel and accompanying figure legend is vague. The text makes it clear that the numbers show the diversity of each library, but it would be helpful to clarify the legend.*

The Figure 2 legend has been revised to clarify the meaning of panel 2D.

*Figure 4/subsection “D. Tite-Seq can measure dissociation constants” – In the figure legend the authors state "Error bars on flow K_D_ values are the same for all data points; they show the average mean squared error computed computed using three replicate measurements for each clone." In the text, they state "Error bars on flow cytometry K_D_ values were computed using the average variance observed in replicate measurements." I have a few problems here. First, neither statement makes totally clear what the authors did/what is shown in Figure 4. Furthermore, MSE usually used when the parametric value of an estimator is known. For the flow data, we don't have parametric values, just three replicates. So, a confidence interval would be more appropriate. Finally, and most troublingly, the authors have elected to present the mean variance/MSE of all three clones rather than the variance (or CI, or whatever) of each clone independently. I can't think of a good reason to do this. Because the WT replicates happened to look great while the two clones didn't, using the mean of all three is misleading. The authors should improve this analysis and then better explain it. Also, note the repeated "computed."*

The revised text now provides clone-by-clone estimates of *K_D_*values and their associated uncertainties.

*Figure 4/Subsection “D. Tite-Seq can measure dissociation constants” – The error estimation procedure for Tite-Seq, based on a single experiment, uses synonymous variants to estimate the standard deviation of each measurement. Variants are binned based on read depth and then a regression is performed to determine the error in each bin. If I understood the procedure, the error is assumed to be wholly dependent on read depth. The read depth vs. signal to noise/error plot (Figure 4) is really noisy, which suggests that the read depth alone does not capture all of the error. Many factors are known to impact experiment-to-experiment variability (e.g. PCR bias, experimenter error, etc.) in these types of experiments. Replication would enable much more robust error estimation, and would considerably strengthen the work.*

The revised text reports the uncertainty of each Tite-Seq-measured *K_D_*value using results from three replicate Tite-Seq experiments.

*Subsection “D. Tite-Seq can measure dissociation constants” – The authors state: "We note that…measurements for the WT scFV are about a factor of 10 larger than the previously measured value of K_D_". I wonder if this is due to the lack of data points at lower concentrations rather than any differences in buffer/etc. Again, a gold-standard, non flow-based assay would help here.*

The reviewer’s intuition seems to have been correct. The improvements to our yeast display assay, as well as a shift downward in the antigen concentration range used, have substantially reduced the discrepancy between our measured WT *K_D_*values (1.9 nM from Tite-Seq and 2.4 nM from flow cytometry) and the previous results reported by Dane Wittrup’s group (1.2 nM, as reported by Gai and Wittrup, 2010).

*Figure 6 – Because of the two-dimensional heatmap it is somewhat difficult to appreciate the relationship between expression and binding. Two separate panels with individual color schemes might be easier to comprehend.*

The new Figure 5—figure supplement 1 show density estimates for the *K_D_*and expression levels of single-point CDR1 and CDR3 mutants. We believe that this more clearly illustrates the typical effect that mutations in these two regions have on affinity and expression.

*Appendix C – The authors say they used custom microarray nucleotides to generate the library, but the nature of the sequences on the array is unclear. A supplementary data file containing the sequences ordered should be included.*

A listing of the sequences in our microarray-synthesized CDR1H and CDR3H libraries is now provided online, along with our analysis code and preprocessed data.

*Appendix F – The authors used the 10% lowest fluorescence values to estimate autofluorescence/background. This is a curious choice, if the library contained mutations to stop codons (see comment above). Stop codons, particularly early ones, virtually guarantee loss of function.*

The revised manuscript uses a more principled analysis approach, which is detailed in Appendix F. In particular, the mean fluorescence of cells at 0M antigen is used to estimate autofluorescence.

[Editors' note: the author responses to the re-review follow.]

*Specific points:*

*1) Please make sure that your deep sequencing data is available on the SRA and relevant computer code is available as supporting file or on a publicly accessible repository. If this is already the case, please clearly indicate in the manuscript where these can be found.*

As described in the Methods, raw data has been deposited on the SRA (BioProject ID PRJNA344711), and both processed data and scripts have been posted at github.com/jbkinney/16_titeseq.

*2) There are a few additional papers that you should consider discussing in the context of prior work:*

*Doolan KM, Colby DW: Conformation-dependent epitopes recognized by prion protein antibodies probed using mutational scanning and deep sequencing. J. Mol. Biol. 2015, 427:328-340.*

*Van Blarcom T, Rossi A, Foletti D, Sundar P, Pitts S, Bee C, Melton Witt J, Melton Z, Hasa-Moreno A, Shaughnessy L, et al.: Precise and efficient antibody epitope determination through library design, yeast display and next-generation sequencing. J. Mol. Biol. 2015, 427:1513-1534.*

We thank the reviewers and editor for this suggestion. These two papers, along with Kowalsky et al. JBC (2015) are now cited in the Introduction as showing how yeast-display-based DMS experiments can be used for mapping the antibody binding epitopes of proteins. We have also cited these papers in the Discussion as examples of the type of experiment that probably does not require full titration curves.

*3) Your approach estimates fairly accurate K_D_ at the cost of more experiments (multiple binds, multiple expression levels). For certain applications where precise K_D_ values are overkill, it may be more efficacious to use the simpler designs like those used in some of the references that you cite – perhaps worth mentioning.*

This is a good point. The revised Discussion section addresses this matter, i.e., that many experiments (such as the epitope mapping studies discussed above) do not require quantitative *K_D_*measurements and that, in such cases, it will often make sense to use a simpler experimental design.

*Also, do you have any comments on the trade-off between sequencing depth and the accuracy of the inferred K_D_? Similarly for the number of sorting bins? Right now it isn't clear how these were chosen. Clearly your choices worked fine, but it would be nice to explain if there was rigorous rationale for choosing these (alternatively, you could just say that exploring the effects of these parameters is interesting for future work).*

We have performed simulations that explore the effect of sequencing depth on the precision of *K_D_*estimates (Figure 4—figure supplement 8). We have not, however, experimentally tested variations in sequencing depth, the number of sorting bins, or the number of different antigen concentrations. The number of bins we used for sorting, as well as the number of antigen concentrations, was chosen in large part for experimental convenience (e.g., the FACS instrument can sort into 4 bins simultaneously, and the number of sorted cells was chosen to enable a full sort to be completed in about 5 hours). These matters are mentioned in the revised Discussion. We also discuss the fact that the precision of our measured *K_D_*values appears to have been limited by the efficiency with which antibody sequences were recovered from sorted cells, and that improvements in the post-sort recovery of such sequences would probably improve the accuracy of Tite-Seq.

*4) The relatively large unsigned error on two variants in Figure 4 (approximately 2 orders of magnitude in K_D_) should be commented on. Why is there such a discrepancy (Subsection “D. Tite-Seq can measure dissociation constants”; Figure 4)?*

The revised Results section now mentions this discrepancy. We do not know the cause of this. The revised text, along with the new Figure 4—figure supplement 3 (see below) notes that the analysis of uncertainty using synonymous mutants reveals a high-variability region of affinity (*K_D_*~1E-7 M) that coincides with these outliers.

*5) Are the mutations outside of the dynamic range (K_D_ > 10 μM) being used in determining correlation coefficients (Figure 4; Figure 4—figure supplement 1; Figure 4—figure supplement 2)?*

Yes. This point is clarified in the revised Figure 4 caption, in the caption for Figure 4—figure supplement 2, and in the main text.

*6) Please quantify the error in precision and accuracy of the method. In particular, the poorer correspondence between replicates in the binding range between 10-1000 nM K_D_ should be mentioned.*

We have included an additional supplemental figure, Figure 4—figure supplement 3, which shows the error estimates for both *K_D_*and E using synonymous mutations. These plots indicate substantially higher noise at ~1E-7 M, and this point is mentioned in the revised main text and figure caption.

*7) Can you quantify error within replicates by inferring K_D_ for synonymous mutations?*

Yes, see the new Figure 4—figure supplement 3.

*8) How big is the benefit of controlling for active surface-displayed protein? While it is clear that yeast surface expression and folded surface displayed protein varies between variants (Burns et al., 2014), it has been shown (yet still surprising!) that surface expression (and proper folding) effects are modest for yeast surface display experiments, particularly for residues that are reasonably surface-exposed (Kowalsky and Whitehead, 2016).*

*Burns, Michael L., et al. Directed evolution of brain-derived neurotrophic factor for improved folding and expression in Saccharomyces cerevisiae. Applied and environmental microbiology 2014, 80:5732-5742.*

*Kowalsky CA, Whitehead TA Determination of binding affinity upon mutation for type I dockerin-cohesin complexes from Clostridium thermocellum and Clostridium cellulolyticum using deep sequencing. PROTEINS 2016 84: 1914-1928*

This is an important point and is now addressed at greater length in the Discussion section. While there has been work (e.g. Kowalsky, 2016) suggesting that the effect of mutations on the expression and specific activity of displayed proteins is modest, other work (Burns et al., 2014) finds that these effects can be quite large. In fact, the magnitude of these effects is likely to vary substantially from protein to protein in a largely unpredictable manner. Our main point, which is now clarified, is that one does not know a priori if these contaminating effects will be present, and that one can guard against them only by assaying full binding curves.